# Extracting rubber tree parameters and estimating carbon storage using airborne LiDAR

**Haoyu Tai**[ID]¹, **Chuangjiang Rao**[ID]²*, **Xia Li**², **Hongen Li**¹, **Chen Li**¹

**1** Yunnan Institute of Water and Hydropower Engineering Investigation, Design Co., Ltd, Kunming, Yunnan, China, **2** Yunnan Institute of Water and Hydropower Engineering Investigation, Design and Research, Kunming, Yunnan, China

* 15911571179@163.com

## Abstract

Facing the dual challenges of global warming and carbon neutrality, forestry carbon sinks play a vital role in achieving carbon neutrality. Rubber plantations, in particular, offer significant ecological and economic co-benefits. However, the efficient and rapid acquisition of data on rubber plantations and the calculation of carbon stock remain key challenges in forestry carbon sink studies. Airborne LiDAR is a powerful tool for forest surveys, yet its inability to directly measure DBH remains a major limitation. This study seeks to address this issue. High-resolution point cloud data were collected, followed by noise removal and ground point classification. Four individual tree segmentation methods were compared, and a linear regression model based on crown diameter parameters was proposed to estimate DBH. The results indicate that the direct point cloud segmentation method achieved the highest accuracy in tree identification. The proposed linear regression model for DBH estimation effectively predicts DBH, enabling precise biomass estimation. The total biomass estimated in the study area was 592,770.57 kg (aboveground biomass: 550,336.17 kg, belowground biomass: 42,434.39 kg), with the corresponding total carbon stock estimated at 278,602.17 kg.

## 1. Introduction

In the context of the global strategy for carbon neutrality, forests, as a major component of terrestrial ecosystems, are crucial in achieving climate goals through their carbon sequestration capabilities [1,2]. Global forest ecosystems are estimated to store substantial amounts of carbon, and even small changes in this storage can significantly influence atmospheric $CO_2$ levels. According to the FAO's 2020 Global Forest Resources Assessment, global forest carbon stocks slightly decreased from 66.8 billion tons in 1990 to 66.2 billion tons in 2020. However, during the same period, forest carbon density (carbon stock per hectare) saw a slight increase, rising from 159 tons per hectare to 163 tons per hectare. Regarding carbon absorption, global

**Data availability statement:** All relevant data are within the manuscript and its Supporting Information files.

**Funding:** This work was financially supported by the National Natural Science Foundation of China (Grant Nos. 62266026); the High Score Special Project: Yunnan Provincial Government Comprehensive Governance Deep Application and Scale-up Industrialization Demonstration Project. (Grant Nos. 89-Y50G31-9001-22/23). The funders had no role in study design, data collection and analysis, decision to publish, or preparation of the manuscript.

**Competing interests:** The authors have declared that no competing interests exist.

forests maintained a roughly balanced net carbon absorption from 2011 to 2020, with an average annual net absorption of approximately −20 million tons of CO2. This balance was the result of the offset between carbon sequestration by forests (−330 million tons) and carbon emissions from deforestation (310 million tons) [3,4]. China has made remarkable progress in forest carbon sequestration. By the end of 2023, the country's forest coverage rate exceeded 25%, and its forest stock had surpassed 20 billion cubic meters. Additionally, the annual carbon sink capacity exceeded 1.2 billion tons of $CO_2$ equivalent, placing China at the forefront globally [5]. With the growing urgency of achieving carbon neutrality, the accurate estimation of carbon stocks has become increasingly crucial in the fight against climate change and the pursuit of carbon neutrality goals. In September 2020, China explicitly set forth its goals for achieving a 'carbon peak' by 2030 and 'carbon neutrality' by 2060. Forestry carbon sequestration has emerged as one of the most potent tools currently available for addressing global climate change and realizing the carbon neutrality objective. It has become a strategic preference for China in its pursuit to reach carbon neutrality by the year 2060.

Forestry carbon sinks involve various measures aimed at enhancing carbon fixation and storage, such as establishing plantations and optimizing forest management. Plantations directly increase carbon absorption, while improved forest management enhances carbon storage efficiency [6]. Forestry carbon sinks play a vital role in achieving carbon neutrality. They not only help mitigate climate change but also provide a range of ecological and economic benefits. As a key strategy for reaching global carbon neutrality goals, the effective utilization of forestry carbon sinks requires precise, efficient, and repeatable measurement, reporting, and verification (MRV) of carbon stocks [7]. This is particularly important in countries like China, where forests contribute to both regional economic development and rural revitalization [8]. Consequently, there is an urgent need to develop carbon sink measurement technologies that are both cost-effective and scientifically robust. To accurately estimate the biomass in forestry carbon sink CCER (Chinese Certified Emission Reduction) projects, several measurement and monitoring technologies have been developed. Common methods for estimating biomass include plot inventory, micrometeorological techniques, and remote sensing [9]. The plot inventory method relies on field surveys to gather forest structural parameters, providing high-accuracy data. However, it is time-consuming, labor-intensive, and costly, making it difficult to cover large areas [10,11]. The micrometeorological method estimates biomass based on climate and microclimate data. It is suitable for long-term monitoring, but it requires complex data and expensive equipment [12]. Remote sensing techniques, such as satellite and laser scanning, offer wide coverage and low cost for large-scale assessments. However, data accuracy may be affected by spatial resolution and atmospheric conditions [13,14].

Rubber trees are a tropical species, indigenous to the towering forests of the Amazon Basin and are the primary source of natural rubber [15]. Introduced to Yunnan Province in China in 1904, rubber trees were cultivated to meet the demands of the then burgeoning industrialization and economic growth. With the rapid industrial

development in tropical regions, rubber trees have been extensively planted across these areas. Currently, plantations of artificial rubber forests are primarily located in tropical regions such as Hainan Island, Xishuangbanna (Yunnan Province), and Guangxi [16]. The economic value of carbon sequestration in artificial rubber forests is primarily manifested in their ability to absorb carbon, thereby reducing greenhouse gas emissions, and generating economic gains through carbon trading markets. The rapid growth of rubber forests makes them an effective source of carbon sequestration, and, in addition to traditional rubber production, they also provide multiple benefits (such as latex and timber, as shown in Fig 1) that support the local economy. Moreover, the ecosystem services provided by artificial rubber forests, such as soil and water conservation and biodiversity protection, contribute to long-term ecological and economic benefits for regional sustainable development [17–20].

Many researchers have also conducted relevant studies on the estimation of tree biomass. For instance, the study by Yin et al. [21] highlighted the importance of accurately estimating regional carbon stocks and emphasized the role of scientific methods in preventing underestimation. Carbon neutrality relies on reducing greenhouse gas emissions and increasing carbon absorption to ultimately balance the concentration of $CO_2$ in the atmosphere, thereby mitigating climate change, protecting the ecological environment, and promoting sustainable development. The research by Adnan Ahmad et al. [22] pointed out that regional carbon emissions related to forest cover change (FCC) and timber harvesting (WH) are crucial for accurately estimating the long-term global carbon balance; Xu Rui et al. utilized the complete harvesting method to collect the average standard timber from rubber trees, delineated sampling squares to gather soil, under-storey vegetation, and litter, followed by weighing and measuring the carbon content of all samples for biomass and carbon stocks calculation [23]. In another study, Guan Limin et al. assessed soil organic carbon stocks by measuring the organic carbon content in rubber plantation forests of varying ages in western Hainan [24]. Roberta Franco Pereira de Queiroz et al. estimated aboveground biomass stocks in savannas using photogrammetric techniques twice, which was documented in both [24] and [25]. Tai Haoyu et al. made significant contributions by constructing a point cloud of plantation forests through laser SLAM technology and conducted preliminary research on the estimation of carbon stocks in small-scale plantation forests [26]. Yifeng Yang et al. developed a canopy relative height model (CRHM) based on UAV LiDAR data, which reflects the height variations of natural grassland vegetation and addresses the large errors present in the canopy height model (CHM) [27]. Er Wang et al. employed multi-source remote sensing and an optimized Least Absolute Shrinkage and Selection Operator (LASSO) variable selection method to improve the accuracy of forest aboveground biomass estimation [28].

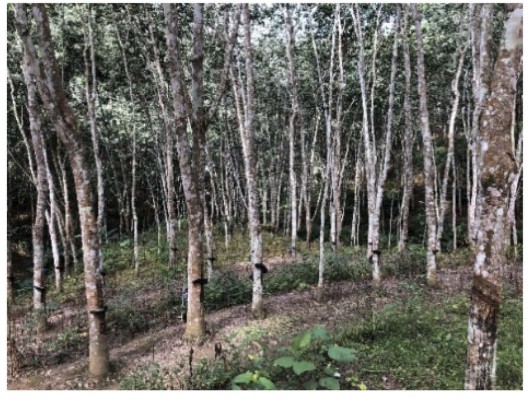
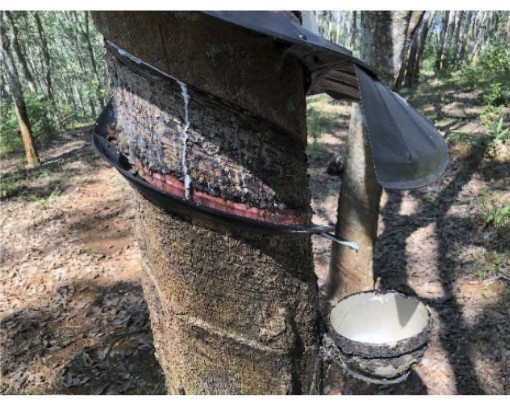

(a)　　　　　　　　　　　　　　(b)

**Fig 1. Artificial rubber plantation and rubber latex.** (a) Artificial Rubber Plantation; (b) Rubber Latex.

However, LiDAR (Light Detection and Ranging) technology has gained significant attention and wide application in forest ecology and carbon sink research. This emerging technique uses laser scanning to accurately capture 3D forest structural data, such as tree height, canopy density, and distribution [29]. As a high-precision remote sensing tool, LiDAR plays a key role in estimating tree biomass, which is essential for carbon sink assessment and monitoring, especially in complex forest environments. With ongoing technological advancements, LiDAR has become an indispensable tool for estimating forest carbon stocks, optimizing carbon trading mechanisms, and addressing climate change challenges [30]. Several studies have demonstrated the potential of LiDAR for estimating forest carbon stocks. For instance, Martin et al. used LiDAR to estimate the biomass of northern shrubs in central and southern Alaska [31]; Widodo Eko Prasetyo et al. employed low-cost backpack LiDAR technology for carbon stock estimation and 3D tree modeling [32]; and Linjing Zhang et al. combined airborne LiDAR with Sentinel-1 and Sentinel-2 time series data for mapping aboveground biomass in semi-arid forests [33].

In biomass estimation, tree height and DBH are the two most closely correlated key parameters. Tree height typically represents a tree's growth stage and its carbon storage potential, while DBH directly influences the tree's cross-sectional area and biomass distribution. However, airborne LiDAR has some limitations in capturing the point clouds of broadleaf trees. During data collection, tree-to-tree occlusion can prevent LiDAR from fully capturing the surface structures of individual trees. Additionally, when the laser signal encounters the canopy, it struggles to penetrate through the tree's internal structure. These challenges make it difficult for LiDAR to directly measure DBH, thereby hindering the accurate estimation of biomass. In terms of tree parameter estimation, Raffaella Brigante et al. used a GIS-based method with high-resolution satellite imagery and shadow analysis to estimate olive tree height [34]. Shaoyi Chen et al. applied airborne LiDAR and a species-specific tree height-diameter model to extract forest structural parameters in the Greater Khingan Mountains [35]. Matthew Guenther et al. estimated DBH using an iPad Pro LiDAR sensor [36]. Kyeongnam Kwon et al. used Mask R-CNN and Bayesian regression OA to estimate DBH from aerial photos [37].

Therefore, the research on the rapid estimation of large-scale tree biomass using airborne 3D laser scanning technology is still in its early stages. This paper aims to evaluate the role of artificial rubber plantations in forestry carbon sequestration and proposes a method to calculate their biomass based on airborne LiDAR point cloud data. The method enables more accurate assessment of the carbon stock in rubber plantations, providing new technical tools for forest carbon sequestration studies and offering a scientific basis for future carbon emission monitoring and management.

## 2. Overview of the study area

The study area is located in the Dongfeng Farm Rubber Plantation, Menglong Town, Jinghong City, Xishuangbanna Dai Autonomous Prefecture, Yunnan Province, China. Yunnan has become the largest rubber plantation region in China. The region experiences a tropical monsoon climate, with an average annual temperature of 21.2°C, an elevation ranging from 600 to 800 meters, and annual rainfall of 1,445.5 mm. The rubber plantations are arranged in a square pattern, with an average row width of 1 meter and an average spacing of 2 meters between plants. The main cultivars planted are RRIM600, GT1, and Yunyan-774 [38]. The project is provided by Yunnan Institute of Water & Hydropower Engineering Investigation, Design Co., Ltd. A section of the rubber plantation was selected for this experiment, where all the rubber trees belong to the same cultivar, as shown in Fig 2 (The picture shows the orthophoto image captured by our own drone).

## 3. Point cloud data acquisition and processing

### 3.1. Principle of UAV-based 3D LiDAR scanning technology

Unmanned aerial vehicle (UAV)-mounted 3D LiDAR scanning technology primarily consists of a laser scanner, Global Navigation Satellite System (GNSS), Inertial Measurement Unit (IMU), UAV platform, and control systems. The operational principle of the system involves the following steps: the UAV-mounted laser emits short laser pulses; these pulses

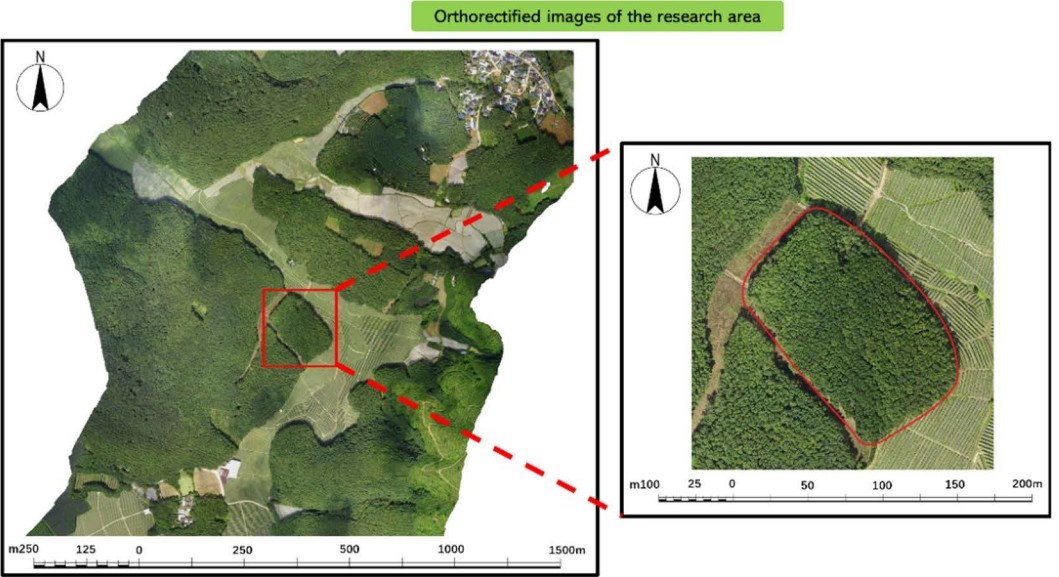

illuminate the ground or target surface; the reflected laser signals are captured by an onboard receiver; and based on the time difference between the emission and reception of the laser pulses, distances are measured by timing the pulses with the speed of light. The principle and workflow are illustrated in Fig 3.

### 3.2. Point cloud data collection

In this study, we used the BB4 rotary-wing UAV (Fig 4a) equipped with the AU20 LiDAR system to collect data from an artificial rubber tree plantation in the study area. The AU20 LiDAR system (Fig 4b) consists of a laser scanning unit, a positioning and attitude determination system, and an external camera. The system parameters are provided in Table 1. The AU20 LiDAR system is known for its strong penetration capabilities, high-precision data acquisition, high operational efficiency, strong adaptability to various environments, and excellent stability, which makes it ideal for capturing three-dimensional tree data.

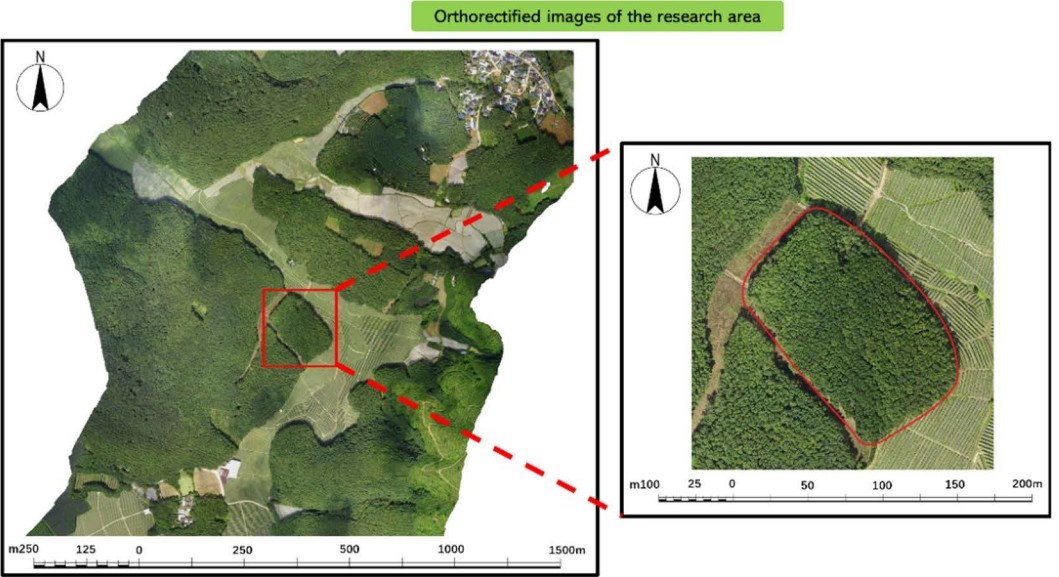

**Fig 2. Overview map of the research area.**

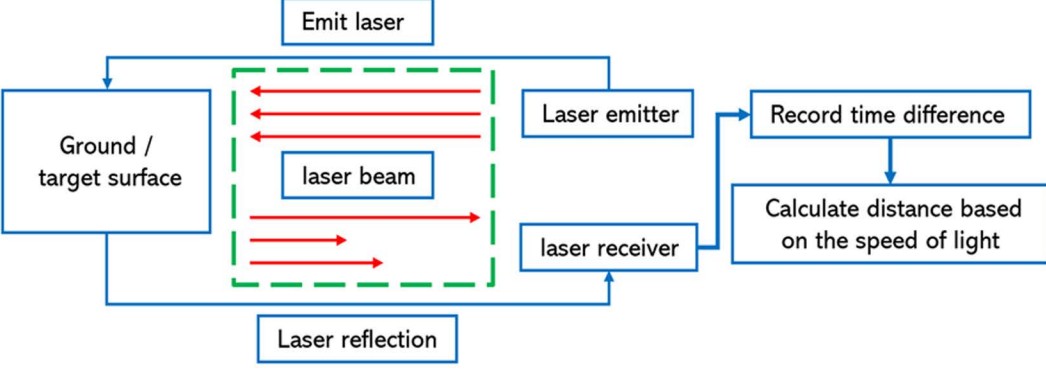

**Fig 3. Schematic diagram of the working principle of UAV-borne 3D laser scanning technology.**

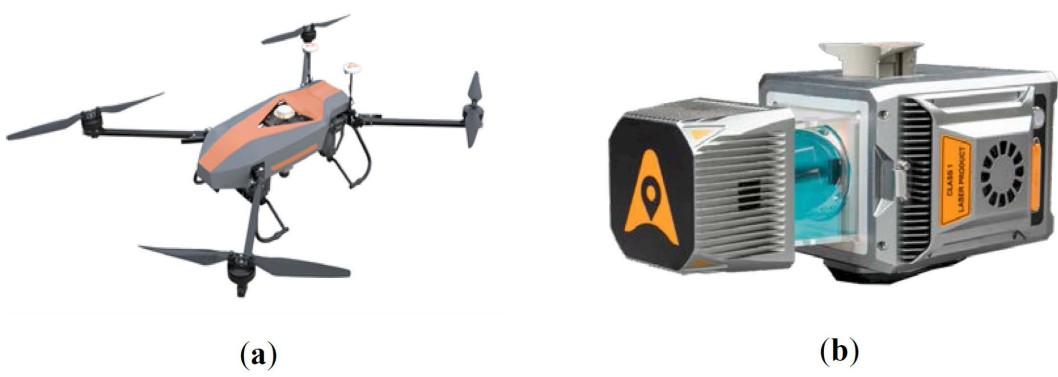

(a) (b)

**Fig 4. UAV-borne LiDAR system.** (a) BB4 quadcopter UAV; (b) AU20 LiDAR.

**Table 1. BB4 Quadcopter UAV with AU20 Lidar technical parameter sheet.**

| BB4 quadcopter drone | | AU20 LiDAR | |
|---|---|---|---|
| Technical Parameter | | Technical Parameter | |
| Empty weight | 10.9 kg | Ranging range | 1.5 ~ 1500m |
| Load weight | 7.0 kg | Maximum emission point frequency | 2 million point/second |
| flight time | 55/45 min (1/5 kg load) | Scanning line speed | 100 lines/second |
| Maximum level flight speed | 14m/s | echo signal | 16 times |
| Ceiling height | 5000 m (Plateau blades) | Ranging accuracy | ±1.5 cm |
| Wind resistance level | Level 6 | Repetitive accuracy | ±0.5 cm |
| Hover accuracy | (Horizontal) 1 cm + 1 ppm | Image resolution | 45 million pixels |
| Wind resistance level | (Vertical) 2 cm + 1 ppm | Postprocessing attitude accuracy | Roll/Pitch: 0.005° |
| working temperature | −10°C ~ +40°C | | Heading: 0.010° |

In the study area, CoPilot flight planning software was used to design flight routes. We employed a variable-height terrain-following flight mode and carried out polygonal route planning over the research area. The key parameters included a relative flight altitude of 300m, a flight direction overlap of 70%, a lateral overlap of 55%, and RTK positioning technology using Qianxun Cors. Since the drone and LiDAR are two separate systems, it is necessary to configure settings such as rotation speed, point leveling, field of view angle, and step angle for the AU20 LiDAR system after completing the route planning. Once these settings are adjusted, a 3D point cloud of the artificial rubber forest in the experimental area can be collected. After the data collection was completed, CoPre point cloud preprocessing software was used to preprocess the point cloud data. The UAV-mounted point cloud preprocessing primarily involves the fusion and calculation of POS (Position and Orientation System) data. This step uses all high-precision navigational information collected by sensors to calculate the precise motion trajectory for the LiDAR system. Additionally, automatic naming, matching, stitching, and coloring of the collected images are performed synchronously. The results of these processes are shown in Fig 5.

### 3.3. Point cloud data processing

To extract tree parameter information relevant to rubber plantations, subsequent data processing of the point cloud is necessary. The first step in data processing involves denoising and filtering the point cloud data to enhance the quality of the rubber plantation point cloud, reduce anomalies within the cloud, and optimize the distribution of point densities. The second step primarily involves point cloud classification (ground point classification), which aims to segregate rubber trees

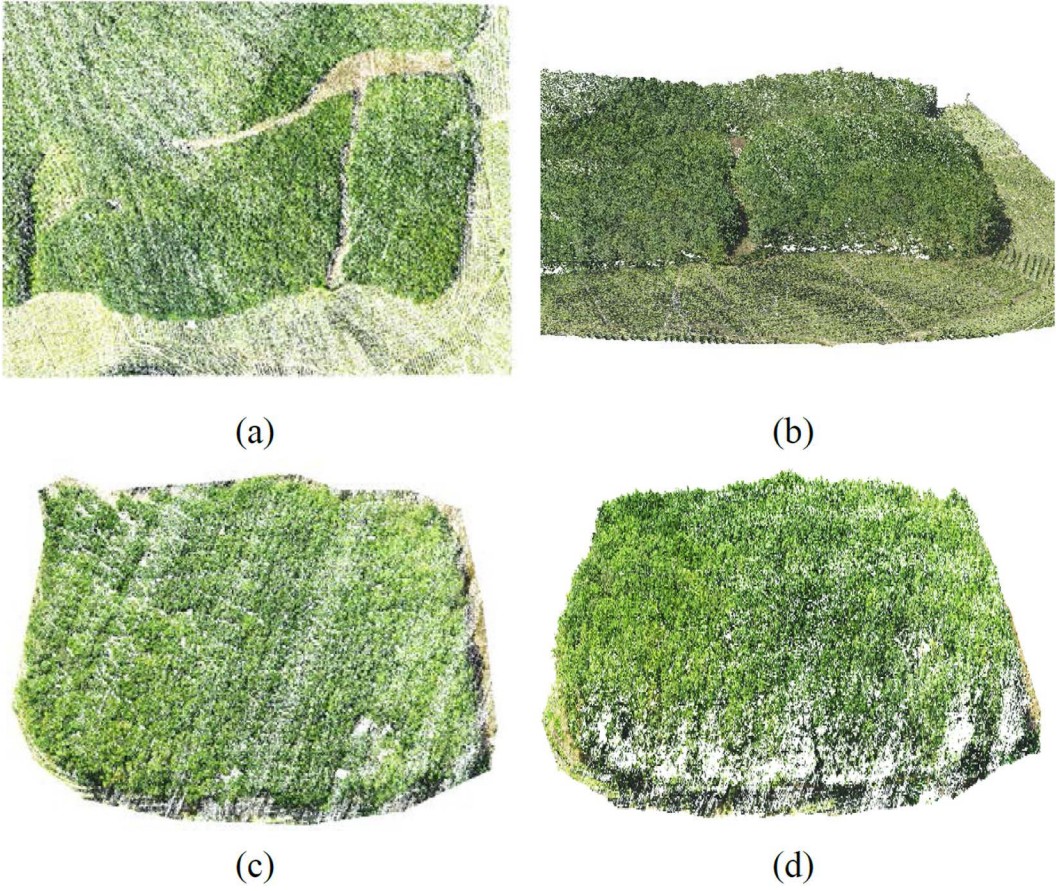

**Fig 5. 3D point clouds of the initial and experimental areas of the artificial rubber forest.** (a) top view of the initial point cloud; (b) top view of the point cloud of the experimental area; (c) iso-metric map in front of the initial point cloud; (d) isometric map in front of the point cloud of the experimental area.

from ground points. The third step entails constructing digital elevation models (DEM), digital surface models (DSM), and canopy height models (CHM) for the experimental area. The data processing work-flow is illustrated in Fig 6, and the processed rubber plantation point cloud is then utilized to extract and analyze the 3D information of rubber trees.

### 3.3.1. Point cloud filtering and denoising.

Due to environmental conditions and equipment precision, the data collection scanning process with LiDAR can be influenced, resulting in the presence of noise and outliers in the three-dimensional point clouds. Noise points, typically arising from sensor errors and environmental interference, appear as random anomalies within the point cloud, which reduce data accuracy and reliability [39]. Outliers, which significantly differ from surrounding points, may be caused by occlusions, sensor inaccuracies, or the edges of the measured objects, potentially leading to shape distortions and registration errors [40]. Therefore, removing these noise and outlier points is a critical data processing step, which helps enhance the quality of the point cloud data and improves the accuracy and effectiveness of subsequent processing. Currently, commonly used point cloud denoising techniques include Gaussian Filtering, Median Filtering, Statistical Filtering, Laplacian of Gaussian (LoG) Filtering, and Bilateral Filtering. We employ Gaussian Filtering as our method for point cloud denoising. Gaussian Filtering [41] is a popular smoothing technique used in the processing of airborne LiDAR point cloud data. The fundamental principle of Gaussian Filtering involves convolution operations using a Gaussian function, where each point and its surrounding points are weighted and averaged to

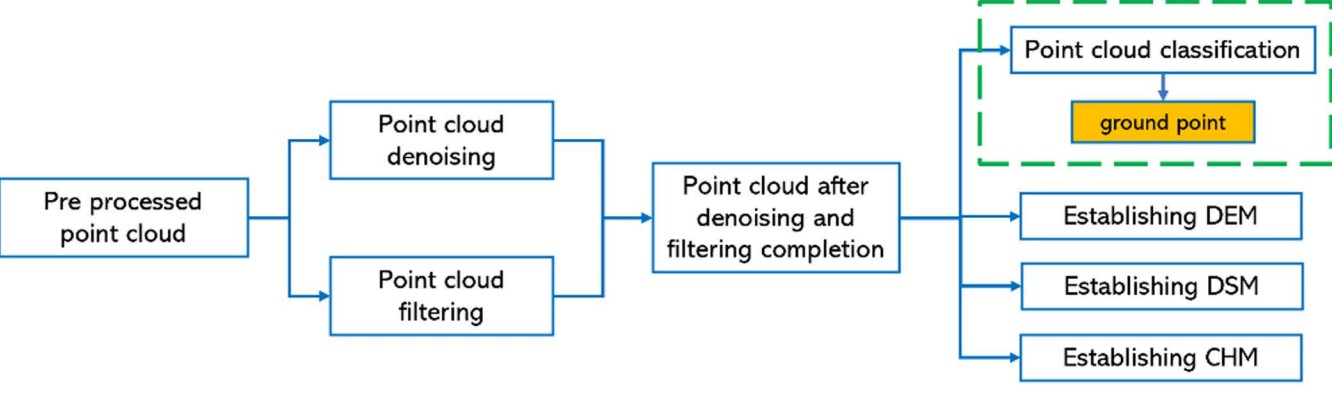

**Fig 6. Flowchart of point cloud data processing for artificial rubber forest.**

effectively smooth the data. In airborne LiDAR point clouds, the value of each point represents its position in three-dimensional space. The Gaussian Filtering algorithm averages these values by weighting them, which helps mitigate local noise and abrupt changes, thereby making the data more continuous and smoother. The results of our Gaussian Filtering denoising process on the rubber forest point clouds are illustrated in Fig 7.

 **3.3.2. Point cloud ground point classification.** Ground point classification of point clouds is crucial for the subsequent construction of Digital Elevation Model (DEM) and for the normalization of point clouds. We employ the Progressive TIN (Triangulated Irregular Network) densification filtering algorithm [42] to classify the ground points in the rubber forest point cloud. The core concept of this algorithm involves gridding the point cloud data, selecting the lowest point in each grid cell as a seed point, and generating a sparse TIN from these seed points. Through iterative processing, the network is progressively densified until all ground points are classified, marking the completion of the iterative densification process [43]. The results of the ground point classification for the rubber forest point cloud are shown in Fig 8.

 **3.3.3. Establishment of DEM, DSM, and CHM.** To extract tree information from rubber tree point clouds, it is essential to generate Digital Elevation Model (DEM), Digital Surface Model (DSM), and Canopy Height Model (CHM) for the study area. DEM is a digital representation of the Earth's surface that records elevation changes at discrete locations. It provides information on the terrain's elevation. DSM represents the Earth's surface along with all features on it, such as terrain, buildings, and trees. In addition to the terrain elevation, the DSM includes the heights of objects on the surface. CHM is used to describe the height of vegetation canopies in forests or areas covered by vegetation. The DEM and DSM are created using the Inverse Distance Weighting (IDW) interpolation method, where nearby points have a greater influence on the interpolation [44]. By overlaying the DEM and DSM, the difference between their elevations is calculated to produce the CHM. The results of these three models are shown in Fig 9.

### 3.4. Single tree segmentation

Tree segmentation refers to the process of partitioning point cloud data into subsets that represent individual trees [45]. By applying tree segmentation to the point clouds of rubber plantations, information on individual rubber trees can be extracted. Currently, tree segmentation methods can be broadly categorized into two main approaches: CHM-based segmentation and point cloud-based segmentation. Among the point cloud-based methods, further distinctions can be made between direct point cloud seg-mentation and seed-point-based segmentation [46]. Deep learning-based indi-vidual tree segmentation has become a significant technological trend, enabling the efficient processing of large-scale three-dimensional point cloud data. This method can accurately identify and segment the spatial structure of individual

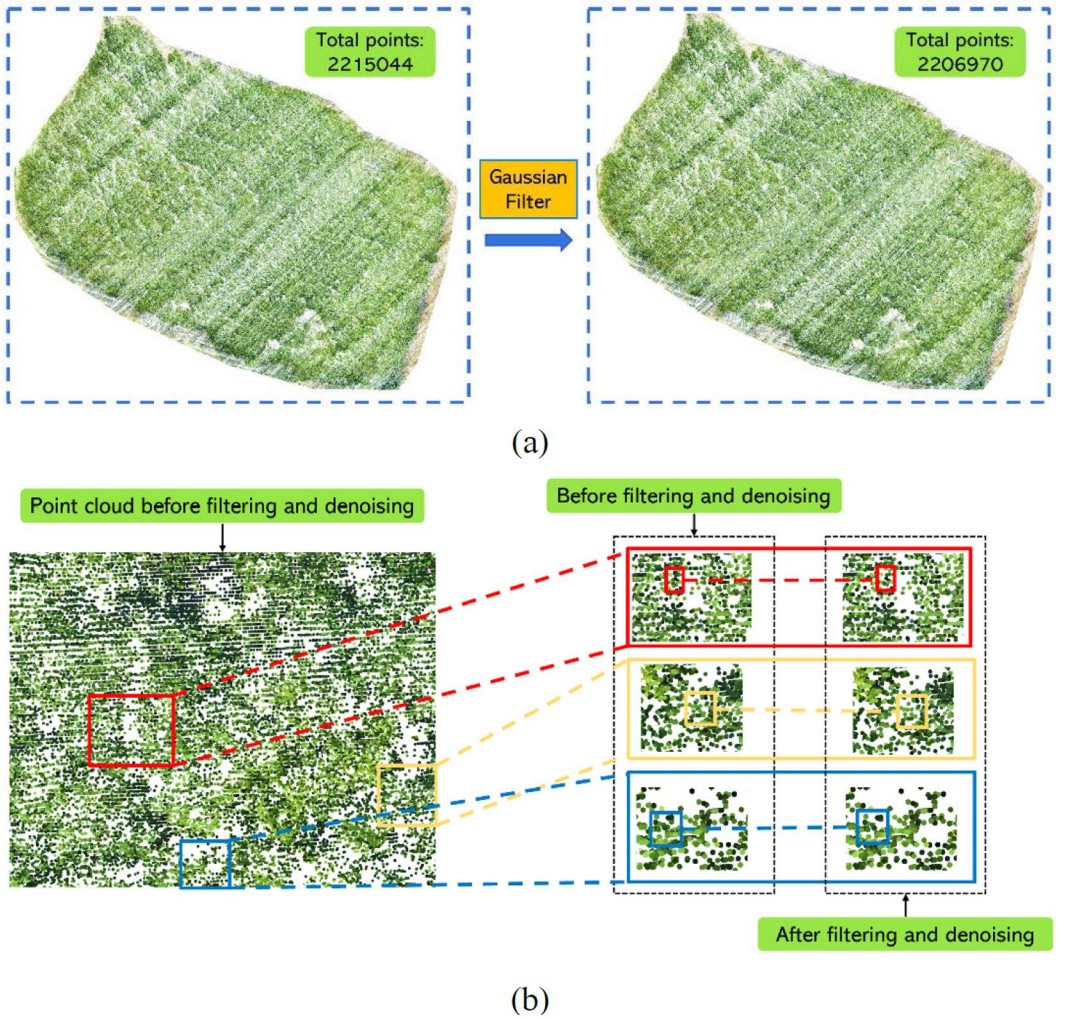

**Fig 7. Filter denoising based on Gaussian filter point cloud.** (a) Number of point clouds after filter denoising; (b) Detail comparison after filter denoising.

trees, overcoming the limitations of traditional approaches, especially in complex environments. In this study, three segmentation methods are compared: CHM-based segmentation, raw point cloud-based segmentation, and seed point-based segmentation. We also explore the application of deep learning techniques for individual tree segmentation. Finally, the most accurate segmentation method is selected to extract tree parameters from rubber tree plantations.

**3.4.1. Single tree segmentation based on CHM.** The principle of single-tree segmentation based on the Canopy Height Model (CHM) is to apply the watershed segmentation algorithm [47]. The primary idea behind using the watershed algorithm is to exploit the topographical or attribute features of the point cloud data by establishing connectivity relationships and constructing a graph that represents the relationships between the points in the cloud. This graph is then used to simulate the flow of water, with the flow paths serving as the boundaries for segmenting the point cloud. The data processing workflow is illustrated in Fig 10. For the seg-mentation of rubber tree point clouds, the following parameters were set: a buffer zone of 50 pixels, a crown height threshold of 0.8 m, a minimum tree height of 7 m, and Gaussian smoothing parameters [Sigma: 0.15, Radius: 9 pixels]. The segmentation results are shown in Fig 12a.

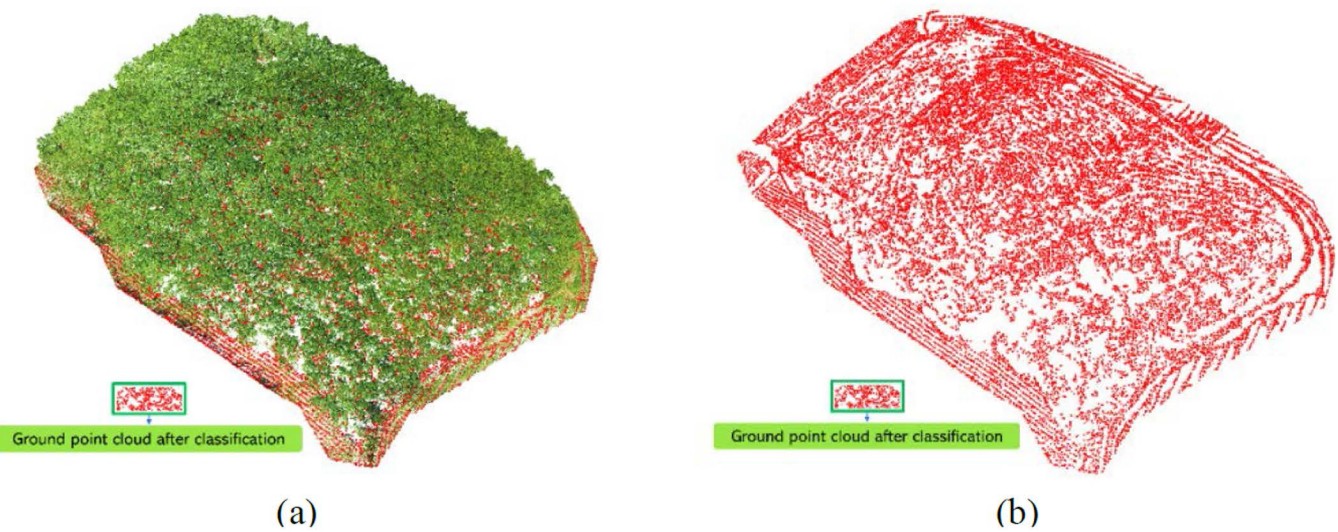

**Fig 8. Post-classification map of point cloud ground points.** (a) point cloud classification results in the experimental area; (b) point cloud of ground points.

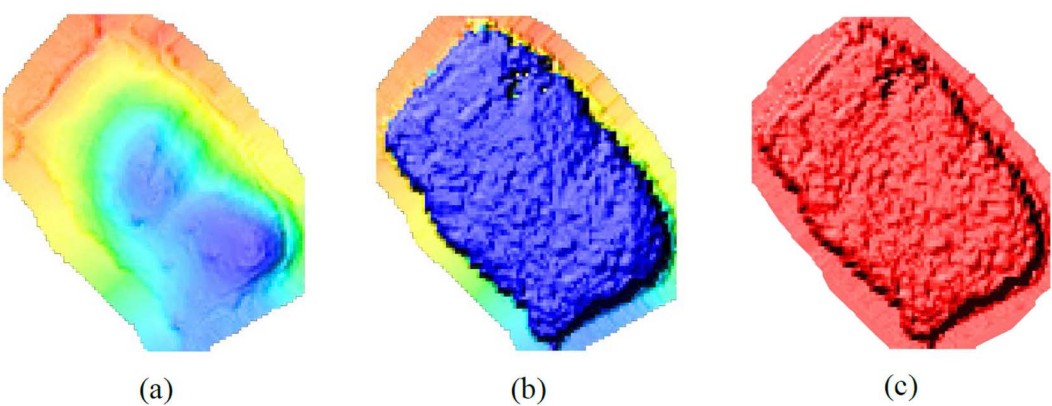

**Fig 9. Shows the processing results of three models.** (a) DEM; (b) DSM; (c) CHM.

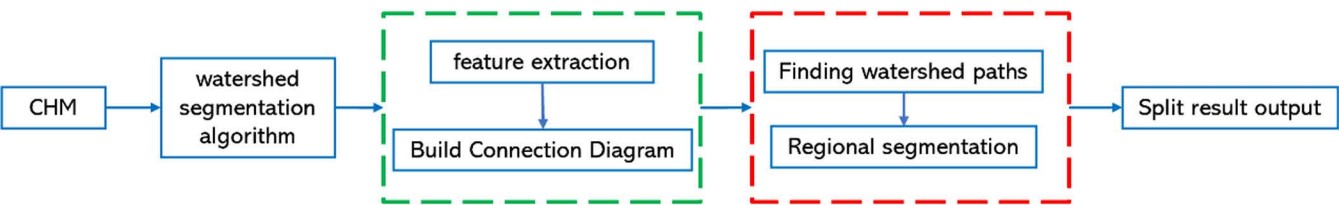

**Fig 10. Flowchart of CHM based single-tree segmentation.**

**3.4.2. Single tree segmentation based on point cloud.** Before performing point cloud-based single tree segmentation, the point cloud data need to be normalized based on ground point data. This normalization process aims to mitigate discrepancies in the point cloud caused by factors such as scale, density, and other variables, ensuring better alignment with the algorithm's requirements and im-proving processing performance. Point cloud segmentation can be categorized into two main approaches: direct point cloud segmentation and seed-based single tree segmentation. The data processing workflow for segmentation is shown in Fig 11. The main approach of the direct point cloud segmentation (PCS) algorithm is to segment individual tree regions from the point cloud data through steps such as elevation value analysis, distance analysis, and clustering. This process allows for the extraction of tree-related attribute information from the point cloud data [48]. Seed point-based individual tree segmentation involves the use of a "seed point" or "seed set" to extend the segmentation process based on either region growth or point-based expansion [49]. The parameter settings for individual tree segmentation using the direct point cloud segmentation algorithm are as follows: grid size of 0.8 m, buffer size of 50 pixels, minimum tree height of 7 m, and Gaussian smoothing parameters [Sigma: 0.75, radius: 5 pixels]. The segmentation result is shown in Fig 12b. For the seed point-based individual tree segmentation, the parameter settings are as follows: ground height of 3 m, minimum tree height of 7 m, and Gaussian smoothing parameters [Sigma: 0.8, radius: 5 pixels]. The result is shown in Fig 12c.

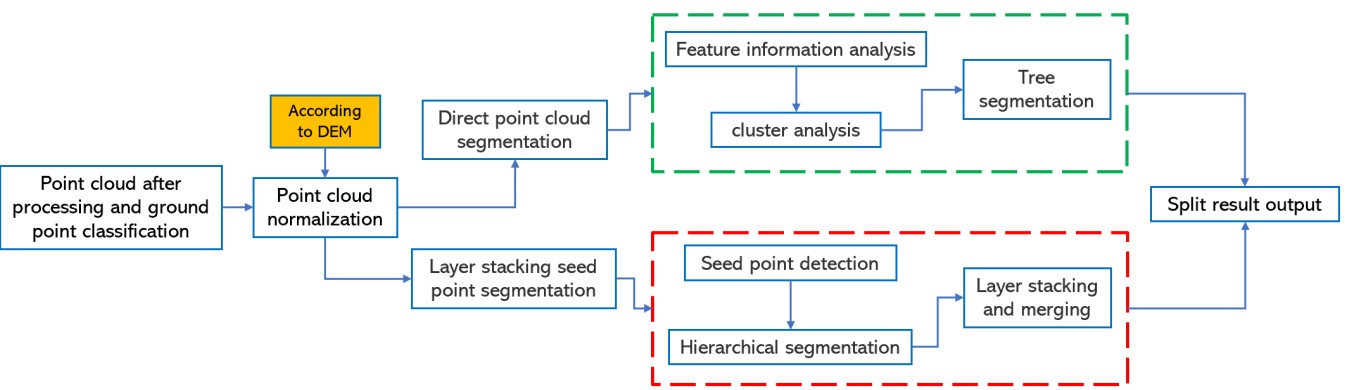

**Fig 11. Flowchart of the point cloud based single-tree segmentation.**

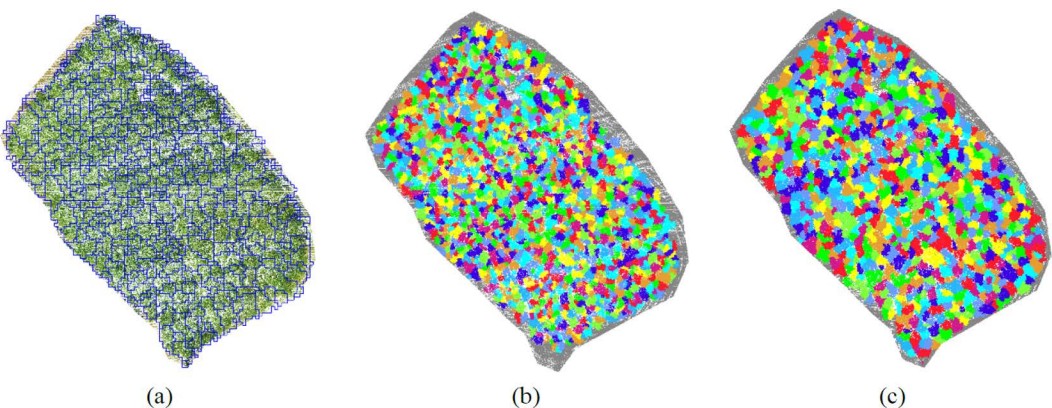

| (a) | (b) | (c) |

**Fig 12. Plots of the results of the three segmentation methods.** (a) CHM based single tree seg-mentation; (b) direct point cloud based single tree segmentation; (c) Seed point-based single tree segmentation.

### 3.4.3. Single tree segmentation based on deep learning.

In recent years, significant advances have been made in using deep learning for individual tree segmentation based on point clouds. This progress is largely due to deep learning's ability to automatically extract complex features from point cloud data, removing the need for manual feature engineering. Techniques such as Convolutional Neural Networks (CNNs), PointNet and its variants (e.g., PointNet++), and Graph Neural Networks (GNNs) have been applied to this task [50]. These methods learn both local and global spatial features of point clouds, enabling the segmentation of individual trees. CNNs [51] typically convert point cloud data into rasterized images or voxel grids before segmentation, whereas PointNet [52] processes raw, unordered point clouds directly, segmenting trees by learning the features of individual points. Deep learning approaches outperform traditional geometry-based methods, especially for dense tree clusters, irregular shapes, or noisy data. They achieve higher segmentation accuracy and robustness, significantly improving the efficiency and precision of forest resource surveys and ecological monitoring.

When processing airborne LiDAR point cloud data, methods based on Convolutional Neural Networks (CNNs) typically require voxelization or gridding preprocessing, which can cause information loss or reduce computational efficiency, especially when dealing with large-scale or sparse point clouds. In contrast, PointNet directly processes raw point clouds, preserving the spatial structure of the data. However, it has limited ability to capture local details. To address this, PointNet++ introduces multi-scale local feature learning, which improves its capability to capture complex structures and hierarchical relationships within the point cloud [53]. As a result, PointNet++ offers better accuracy and robustness, particularly for tasks such as segmenting complex scenes and dense vegetation. In this study, we explore the application of the PointNet++ model for individual tree segmentation.

In order to train the model, it is essential to label the points in the point cloud data. Seventeen rubber trees were manually segmented from airborne LiDAR point cloud data and categorized into two classes: canopy points and ground points, as shown in Fig 13a. To improve the model's adaptability to various scenes, data augmentation techniques, including rotation and translation, were applied. The model extracts local features through the Set Abstraction (SA) layer, recovers detailed information via the Feature Propagation (FP) layer, and performs classification or regression using the fully connected layer. During training, a cross-entropy loss function and data augmentation methods were used to enhance the model's robustness, while an optimizer was employed for parameter updates. During the inference phase, segmentation predictions are made by inputting the preprocessed point cloud data, followed by post-processing to improve segmentation accuracy. The segmentation results are shown in Fig 13b.

As shown in Fig 13c, the individual tree segmentation results from the PointNet++ deep learning model presented several issues, such as under-segmentation, over-segmentation, and failure to segment. We identified the following potential causes for these issues: First, the quality of airborne LiDAR point cloud data may have affected the precise delineation of tree boundaries. Second, the complexity of tree structures, including morphological variations and occlusions, made segmentation more difficult. Additionally, insufficient or unbalanced training data impaired the model's ability to process point clouds effectively, exacerbating segmentation errors. Finally, incomplete data preprocessing, such as inadequate noise removal, may have contributed to the segmentation issues. Consequently, we decided not to use the results from the individual tree segmentation based on the PointNet++ model.

### 3.4.4. Precision analysis of three segmentation methods.

After performing point cloud segmentation on the rubber plantation using three different segmentation methods, the corresponding number of trees segmented by each method was obtained. To compare the segmentation accuracy of the three methods, a field survey was conducted to collect tree count data for the experimental area. The field survey method, commonly used in forestry investigations, involves systematic biological surveys and resource measurements within pre-selected small-scale plots, aimed at ac-quiring comprehensive information on the characteristics of the entire study area [54]. In this study, square plots measuring 24 m × 24 m were used for data collection, with a total of five plots established. The average tree density per plot was 32 trees, as shown in Fig 14. The total area of the experimental region was 21,999.874 m², and based on the field survey method, the total number of rubber trees in the study area was estimated to be 1,222.

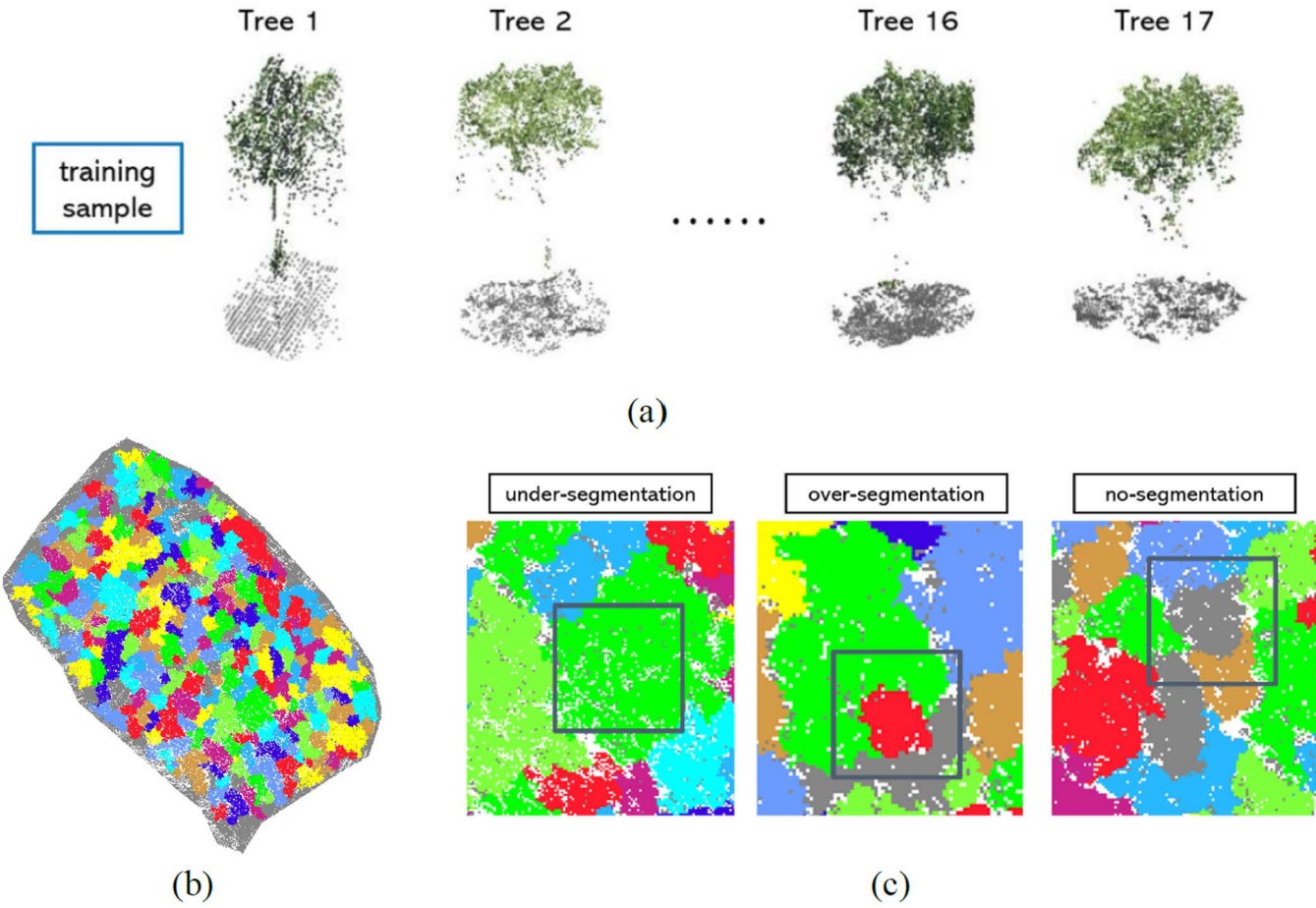

**Fig 13. Single tree segmentation based on deep learning.** (a) training samples; (b) Segmentation results; (c) Segmentation problem.

The tree count estimated by the sampling plot survey method was used as the ground truth. The accuracy of the three segmentation methods was assessed by calcu-lating recall (r), precision (p), and F-score (F) [55]. The formulas for calcu-lating r, p, and F are as follows:

$$r = \frac{TP}{TP+FN}; \quad p = \frac{TP}{TP+FP}; \quad F = 2 \times \frac{r \times p}{r+p}$$

(1)

In the formula, TP denotes True Positive, FN denotes False Negative, and FP denotes False Positive. TP, FN, and FP correspond to the numbers of trees that are correctly segmented, under-segmented, and over-segmented, respectively. The segmentation ac-curacies of the three methods are summarized in Table 2.

When selecting a segmentation method, the F-score is commonly used as an indicator of segmentation accuracy. An F-score close to 1 indicates high accuracy, with both the correlation coefficient (r) and p-value being elevated. In contrast, an F-score near 0 suggests poor segmentation accuracy, with one or both of the r-value and p-value being low. As shown in Table 2, the F-scores for the segmentation methods based on the Canopy Height Model (CHM), direct point cloud, and seed point-based approaches are 0.24, 0.786, and 0.651, respectively. Therefore, the direct point cloud segmentation method, which yields the highest F-score, was chosen for parameter extraction of rubber trees.

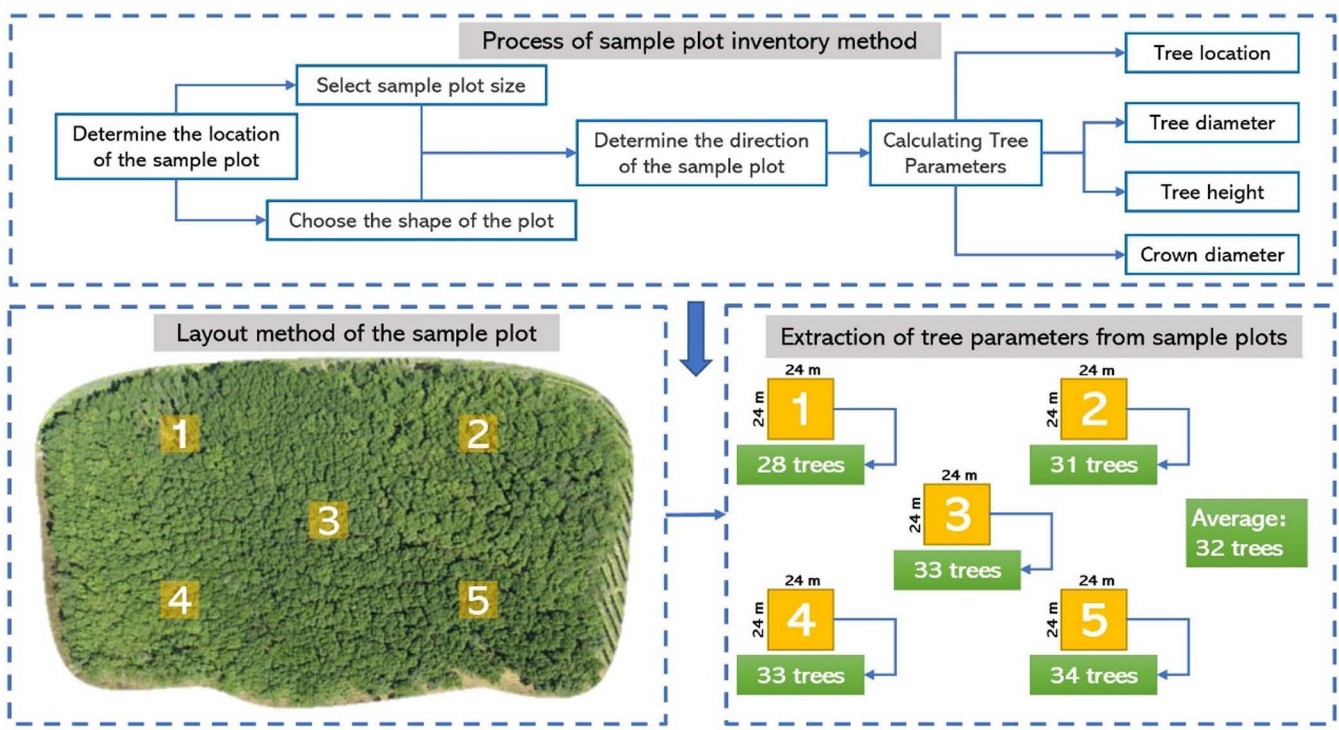

**Fig 14. The process of the sampling plot survey method.**

**Table 2. Segmentation accuracy statistics of the three segmentation methods.**

| Segmentation method | Total number of trees | Total amount of segmentation | TP | FN | FP | r | p | F |
|---|---|---|---|---|---|---|---|---|
| CHM-based Single Tree Segmentation | 1222 (tree) | 392 (tree) | 197 | 862 | 388 | 0.186 | 0.337 | 0.240 |
| Direct Point Cloud- based Single Tree Segmentation | 1222 (tree) | 1320 (tree) | 978 | 240 | 291 | 0.803 | 0.771 | 0.786 |
| Seed-based Single Tree Segmentation | 1222 (tree) | 801 (tree) | 721 | 514 | 259 | 0.584 | 0.736 | 0.651 |

## 4. Parameter extraction and biomass estimation

### 4.1. Rubber tree parameter extraction

The parameters of rubber trees typically include diameter at breast height (DBH), tree height, and crown size. These parameters provide essential data for the accurate estimation of tree biomass. Among the three point cloud segmentation methods, the direct point cloud segmentation method produced the most optimal results. Therefore, point cloud data obtained from single-tree segmentation using this method was selected for the extraction of rubber tree parameters. In this study, Lidar360 software was employed to extract the rubber tree parameters from the single-tree segmentation results, as shown in Table 3.

### 4.2. Accuracy analysis of parameter extraction

To validate the accuracy of the rubber tree parameters extracted by the Lidar360 software, field survey data obtained through plot-based sampling were used as theo-retical reference values. Precision analysis was conducted on the extracted parameters, including tree height, crown diameter, north-south crown diameter, and east-west crown diameter.

**Table 3. Information table of rubber tree parameters extracted based on Lidar360 processing software.**

| Tree ID | X | Y | TH (m) | DBH (m) | CD (m) | N-S CD (m) | E-W CD (m) |
|---|---|---|---|---|---|---|---|
| 1 | …761.24 | …2778.26 | 20.5 | 0 | 5.37 | 6.71 | 4.03 |
| 2 | …770.58 | …2772.47 | 20.9 | 0 | 7.11 | 5.12 | 9.11 |
| 3 | …754.17 | …2779.50 | 21.1 | 0 | 4.27 | 4.23 | 4.31 |
| 4 | …751.66 | …2778.80 | 21.7 | 0 | 6.14 | 5.41 | 6.87 |
| 5 | …764.31 | …2773.33 | 22.1 | 0 | 6.96 | 7.42 | 6.51 |
| … | … | … | … | … | … | … | … |
| 1316 | …801.68 | …2605.75 | 19 | 0 | 7.15 | 7.93 | 6.38 |
| 1317 | …805.78 | …2598.50 | 18.3 | 0 | 8.19 | 7.22 | 9.17 |
| 1318 | …802.93 | …2602.01 | 19 | 0 | 5.91 | 6.74 | 5.08 |
| 1319 | …800.90 | …2596.75 | 21.2 | 0 | 5.96 | 6.62 | 5.31 |
| 1320 | …795.93 | …2596.25 | 20.1 | 0 | 6.85 | 7.34 | 6.36 |

\* Note: Tree Height (TH); Diameter at Breast Height (DBH); Crown Diameter (CD); North-South Crown Diameter (N-S CD); East-West Crown Diameter (E-W CD).

A total of 50 rubber trees were randomly selected, and their corresponding extracted parameters were matched with the coordinates obtained from the field survey data. The results of the accuracy analysis are presented in Tables 4 and 5.

In tree parameter acquisition, controlling measurement errors is crucial. The measurement error for tree height is generally within 1–2%, while the error for crown diameter is typically within 2–5%. Precision analysis revealed that the maximum measurement error for tree height was 0.3 meters, falling within an acceptable margin of error. For crown diameter, the average maximum error was 0.64 meters, with the maximum errors in the north-south and east-west directions reaching 0.96 and 0.23 meters, respectively. Notably, the error in the east-west crown diameter remained within the acceptable error margin.

## 4.3. Linear regression model for DBH

The accuracy analysis reveals that while we successfully obtained parameters such as the tree height, north-south crown diameter, and east-west crown diameter for the rubber trees, we were unable to acquire the DBH data. Tree height and DBH are key parameters that are strongly correlated in biomass estimation for rubber trees. Tree height typically indicates

**Table 4. The accuracy analysis table of rubber tree height and average crown diameter.**

| Tree ID | Tree Height (m) | | | Average Crown Diameter (m) | | |
|---|---|---|---|---|---|---|
| | Theoretical value | Actual value | Error | Theoretical value | Actual value | Error |
| 1 | 16.7 | 16.8 | −0.1 | 6.50 | 6.16 | 0.34 |
| 2 | 17.9 | 18.1 | −0.2 | 5.68 | 5.33 | 0.36 |
| 3 | 17.8 | 17.9 | −0.1 | 5.84 | 5.94 | −0.10 |
| 4 | 17.6 | 17.8 | −0.2 | 6.37 | 6.85 | −0.48 |
| 5 | 19.2 | 19.0 | 0.2 | 5.94 | 6.11 | −0.17 |
| … | … | … | … | … | … | … |
| 46 | 19.1 | 19.3 | −0.2 | 6.06 | 6.12 | −0.06 |
| 47 | 17.9 | 18.2 | −0.2 | 5.59 | 6.08 | −0.49 |
| 48 | 20.5 | 20.5 | −0.1 | 5.89 | 6.23 | −0.35 |
| 49 | 20.0 | 20.3 | −0.3 | 6.06 | 6.34 | −0.29 |
| 50 | 18.2 | 18.0 | 0.3 | 6.30 | 6.80 | −0.50 |

**Table 5. The accuracy analysis table of the north-south and east-west crown diameters of rubber trees.**

| Tree ID | North South Crown Diameter (m) | | | East West Crown Diameter (m) | | |
|---|---|---|---|---|---|---|
| | Theoretical value | Actual value | Error | Theoretical value | Actual value | Error |
| 1 | 6.33 | 5.78 | 0.55 | 6.67 | 6.54 | 0.13 |
| 2 | 5.12 | 4.34 | 0.78 | 6.24 | 6.31 | −0.07 |
| 3 | 5.91 | 6.04 | −0.13 | 5.77 | 5.83 | −0.06 |
| 4 | 6.49 | 7.38 | −0.89 | 6.25 | 6.32 | −0.07 |
| 5 | 5.87 | 6.35 | −0.48 | 6.01 | 5.86 | 0.15 |
| … | … | … | … | … | … | … |
| 46 | 6.14 | 6.07 | 0.07 | 5.98 | 6.17 | −0.19 |
| 47 | 5.42 | 6.31 | −0.89 | 5.75 | 5.84 | −0.09 |
| 48 | 5.87 | 6.54 | −0.67 | 5.9 | 5.92 | −0.02 |
| 49 | 5.83 | 6.35 | −0.52 | 6.28 | 6.33 | −0.05 |
| 50 | 6.17 | 7.09 | −0.92 | 6.43 | 6.51 | −0.08 |

the growth stage and carbon storage capacity of the rubber tree, while DBH directly influences the tree's cross-sectional area and biomass distribution. However, airborne LiDAR has limitations when acquiring point clouds of broadleaf trees. First, tree occlusion can prevent LiDAR from fully capturing the surface structure of all trees, especially in dense canopy layers. Second, when the laser signal encounters the canopy surface, it is often reflected or scattered, hindering its ability to penetrate the internal structure of the tree. These challenges make it difficult for LiDAR to accurately measure DBH, which in turn complicates biomass estimation.

To accurately estimate the breast height diameter (BHD) of rubber trees, we propose constructing a linear regression model that incorporates the parameters of the east-west crown diameter, north-south crown diameter, and average crown diameter. Common linear regression models include power function models, exponential function models, simple linear models, and quadratic models. These models will be developed by using different parameters as independent variables (x) and BHD as the dependent variable (y). The specific regression model equations are presented in Table 6.

We selected five square plots, and within each plot, two groups of rubber trees were randomly chosen, with each group consisting of 10 trees. One group was used for experimental data analysis, and the other was reserved for validation of the results. The following parameters were measured at a height of 1.3 meters above the ground for the 20 selected rubber trees: DBH, north-south crown diameter, east-west crown diameter, and average crown diameter. The data for the 10 experimental trees are presented in Table 7.

Based on the linear regression model described above and the measured data, we established a linear regression relationship by using the sample parameters of E-W CD, N-S CD, and A CD as independent variables, and the DBH as the dependent variable. The fitting results are shown in Table 8, and the fitted curve is depicted in Fig 15.

After obtaining the linear regression model for estimating the diameter at breast height (DBH) based on the parameters of E-W CD, N-S CD, and A CD, we validated the predictive capability and accuracy of the model using the measured data from another 10 rubber trees. The estimation results of the E-W CD and DBH regression model are presented in Table

**Table 6. Expression of linear regression model.**

| Linear regression model | Expression |
|---|---|
| Power function model | $y_1 = ax^b$ |
| Exponential function model | $y_2 = ae^{bx}$ |
| One variable linear function model | $y_3 = ax + b$ |
| One variable quadratic function model | $y_4 = ax^2 + bx + c$ |

**Table 7. Actual measured parameters of DBH, north-south crown diameter, east-west crown diameter, and average crown diameter.**

| TreeID | DBH(m) | E-W CD(m) | N-S CD(m) | A CD(m) |
|---|---|---|---|---|
| 1 | 0.274 | 5.52 | 4.68 | 5.10 |
| 2 | 0.301 | 7.33 | 7.06 | 7.20 |
| 3 | 0.255 | 5.59 | 5.24 | 5.42 |
| 4 | 0.297 | 6.15 | 6.00 | 6.08 |
| 5 | 0.314 | 6.78 | 4.97 | 5.88 |
| 6 | 0.282 | 6.20 | 5.74 | 5.97 |
| 7 | 0.310 | 6.60 | 5.67 | 6.14 |
| 8 | 0.267 | 5.77 | 5.63 | 5.70 |
| 9 | 0.279 | 5.98 | 6.25 | 6.12 |
| 10 | 0.306 | 6.82 | 6.33 | 6.58 |

\* Note: E-W CD is east-west crown diameter; N-S CD is north-south crown diameter; A CD is average crown diameter.

**Table 8. Expression of linear regression model.**

| E-W CD(m) | N-S CD(m) | A CD(m) |
|---|---|---|
| $y_1 = 0.09350x^{0.61391}$ | $y_1 = 0.21531x^{0.16761}$ | $y_1 = 0.13141x^{0.43862}$ |
| $y_2 = 0.15881e^{0.09492x}$ | $y_2 = 0.24312e^{0.02971x}$ | $y_2 = 0.18940e^{0.06981x}$ |
| $y_3 = 0.01158x + 0.2156$ | $y_3 = 0.00851x + 0.23942$ | $y_3 = 0.02081x + 0.16331$ |
| $y_4 = -0.02111x^2 + 0.29722x - 0.73823$ | $y_4 = 0.00331x^2 - 0.03031x + 0.35111$ | $y_4 = -0.01112x^2 + 0.15751x - 0.25342$ |

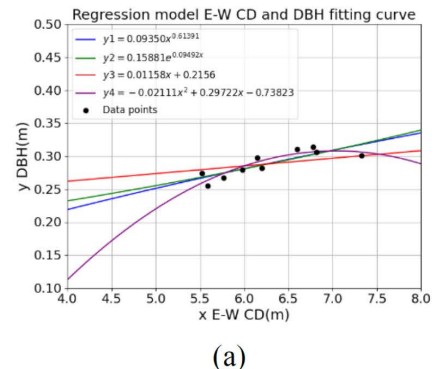
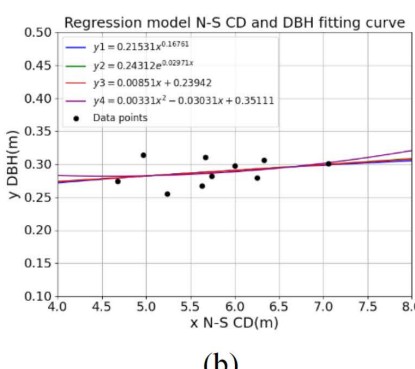
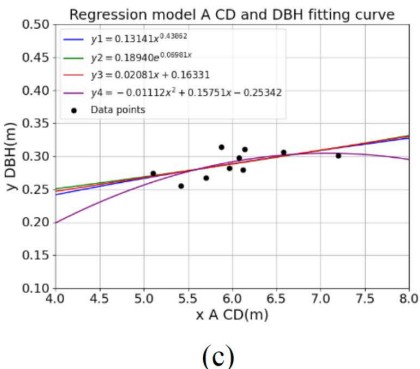

(a) (b) (c)

**Fig 15. Shows the fitting curves of various regression models.** (a) the fitting curve of the DBH regression model estimated by E-W CD; (b) The fitting curve of the DBH regression model estimated by N-S CD; (c) A CD estimates the fitting curve of the DBH regression model.

9, the N-S CD and DBH regression model estimation results are shown in Table 10, and the A CD and DBH regression model estimation results are shown in Table 11.

A linear regression model was developed to estimate the DBH using E-W CD, N-S CD, and A CD parameters. The model was validated by estimating DBH for another set of rubber trees, followed by assessing the maximum, minimum, average errors, and the Root Mean Squared Error (RMSE). The RMSE quantifies the statistical error between the measured DBH and the estimated DBH, with a lower RMSE indicating higher accuracy. The maximum, minimum, average errors, and RMSE values for the DBH regression models of each parameter are provided in Table 12. Our primary focus

**Table 9. Estimated DBH values using E-W CD and DBH regression models.**

| Tree ID | Measured values DBH (m) | Measured values E-W CD (m) | 4 types of linear regression models | | | |
|---|---|---|---|---|---|---|
| | | | $y_1$ | $y_2$ | $y_3$ | $y_4$ |
| 1 | 0.281 | 5.32 | 0.261 | 0.263 | 0.277 | 0.246 |
| 2 | 0.331 | 6.50 | 0.295 | 0.294 | 0.291 | 0.302 |
| 3 | 0.287 | 5.42 | 0.264 | 0.266 | 0.279 | 0.253 |
| 4 | 0.327 | 7.36 | 0.318 | 0.319 | 0.301 | 0.306 |
| 5 | 0.318 | 6.34 | 0.291 | 0.290 | 0.289 | 0.298 |
| 6 | 0.332 | 6.81 | 0.304 | 0.303 | 0.294 | 0.307 |
| 7 | 0.294 | 5.39 | 0.263 | 0.265 | 0.278 | 0.251 |
| 8 | 0.276 | 6.43 | 0.293 | 0.292 | 0.291 | 0.300 |
| 9 | 0.306 | 6.18 | 0.286 | 0.286 | 0.287 | 0.292 |
| 10 | 0.274 | 5.74 | 0.273 | 0.274 | 0.281 | 0.272 |

**Table 10. Estimated DBH values using N-S CD and DBH regression models.**

| Tree ID | measured values DBH(m) | measured values N-S CD(m) | 4 types of linear regression models | | | |
|---|---|---|---|---|---|---|
| | | | $y_1$ | $y_2$ | $y_3$ | $y_4$ |
| 1 | 0.281 | 5.24 | 0.284 | 0.284 | 0.284 | 0.283 |
| 2 | 0.331 | 6.87 | 0.297 | 0.298 | 0.298 | 0.299 |
| 3 | 0.287 | 5.42 | 0.286 | 0.286 | 0.286 | 0.284 |
| 4 | 0.327 | 6.99 | 0.298 | 0.299 | 0.299 | 0.301 |
| 5 | 0.318 | 5.62 | 0.288 | 0.287 | 0.287 | 0.285 |
| 6 | 0.332 | 6.74 | 0.297 | 0.297 | 0.297 | 0.297 |
| 7 | 0.294 | 4.83 | 0.280 | 0.281 | 0.281 | 0.282 |
| 8 | 0.276 | 6.77 | 0.297 | 0.297 | 0.297 | 0.298 |
| 9 | 0.306 | 5.64 | 0.288 | 0.288 | 0.287 | 0.286 |
| 10 | 0.274 | 4.95 | 0.282 | 0.282 | 0.282 | 0.282 |

**Table 11. Estimated DBH values using A CD and DBH regression models.**

| Tree ID | measured values DBH(m) | measured values A CD(m) | 4 types of linear regression models | | | |
|---|---|---|---|---|---|---|
| | | | $y_1$ | $y_2$ | $y_3$ | $y_4$ |
| 1 | 0.281 | 5.28 | 0.273 | 0.274 | 0.273 | 0.268 |
| 2 | 0.331 | 6.69 | 0.303 | 0.302 | 0.303 | 0.303 |
| 3 | 0.287 | 5.42 | 0.276 | 0.277 | 0.276 | 0.274 |
| 4 | 0.327 | 7.18 | 0.312 | 0.313 | 0.313 | 0.304 |
| 5 | 0.318 | 5.98 | 0.288 | 0.288 | 0.288 | 0.291 |
| 6 | 0.332 | 6.78 | 0.304 | 0.304 | 0.304 | 0.303 |
| 7 | 0.294 | 5.11 | 0.269 | 0.271 | 0.270 | 0.261 |
| 8 | 0.276 | 6.60 | 0.301 | 0.300 | 0.301 | 0.302 |
| 9 | 0.306 | 5.91 | 0.287 | 0.286 | 0.286 | 0.289 |
| 10 | 0.274 | 5.35 | 0.274 | 0.275 | 0.275 | 0.271 |

was on the average error and RMSE, as these two metrics give a comprehensive assessment of the model's performance. Smaller values of these metrics indicate lower model errors and better model performance.

For the regression model based on E-W CD, both the average error and RMSE of y3 were low, 0.0202 and 0.0235, respectively. Among all models for y1, y2, y3, and y4, y3 demonstrated the best performance with the smallest error and the lowest RMSE. For the regression model based on N-S CD, the error indicators for y1, y2, y3, and y4 were very similar, with y2 showing slightly better performance, with an average error of 0.0191 and an RMSE of 0.0225. For the regression model using A CD, the average error and RMSE for y2 were optimal, 0.0186 and 0.02092, respectively. However, overall, the performance was slightly worse than the model based on N-S CD. Based on this analysis, N-S CD was identified as the optimal independent variable, as it performed well across all indicators, especially with relatively small RMSE and average errors. However, in terms of parameter extraction, neither N-S CD nor A CD met the required accuracy. Only the E-W CD parameter satisfied the necessary extraction conditions. Therefore, the regression model using E-W CD, given by y3 = 0.01158x + 0.2156, was selected for further research.

## 4.4. Estimation of carbon stock in rubber trees

Estimating the carbon stock of trees is crucial in forestry carbon sequestration pro-jects and forms the foundation of the carbon trading market [56]. Common methods for estimating tree carbon stock include the whole-tree biomass equation, remote sensing approaches, and tree modeling techniques. In this study, we estimate the carbon stock in rubber plantations by extracting tree height and DBH parameters and employing the volume-based biomass method. The volume-based biomass method estimates the biomass of individual trees by measuring parameters such as tree height and DBH, and applying species-specific allometric equations. The carbon stock of each tree is derived by multiplying the estimated biomass by the tree's carbon content. The total carbon stock of the stand is obtained by summing the carbon stock of all trees. The Allometric Growth Equation is a mathematical model used to describe growth patterns in organisms. It is commonly applied to investigate the relationships between the dimensions and biomass of different parts of an organism and to assess its growth potential [57]. According to the Allometric Growth Equation for dominant tree species outlined in the "Guidelines for Carbon Sequestration Measurement and Monitoring in Afforestation Projects" [58], the equation for rubber tree species is used to calculate the aboveground and belowground biomass in artificial rubber plantations within the study area. The calculation formula is shown in Table 13.

The aboveground and belowground biomass of rubber trees can be calculated using the allometric growth equation for rubber trees. By multiplying the biomass with the carbon content, the carbon stock of the rubber tree is determined.

**Table 12. Maximum, minimum, mean error, and root mean square error of DBH estimation regression model for each parameter.**

| Independent variable | Regression model | Max error | Min error | Average error | RMSE |
|---|---|---|---|---|---|
| E-W CD | $y_1$ | 0.036 | 0.001 | 0.0212 | 0.0234 |
| | $y_2$ | 0.037 | 0.002 | 0.0206 | 0.0236 |
| | $y_3$ | 0.040 | 0.004 | 0.0202 | 0.0235 |
| | $y_4$ | 0.043 | 0.002 | 0.0247 | 0.0270 |
| N-S CD | $y_1$ | 0.035 | 0.001 | 0.0193 | 0.0227 |
| | $y_2$ | 0.035 | 0.001 | 0.0191 | 0.0225 |
| | $y_3$ | 0.035 | 0.001 | 0.0192 | 0.0226 |
| | $y_4$ | 0.035 | 0.002 | 0.0193 | 0.0226 |
| A CD | $y_1$ | 0.030 | 0.000 | 0.0189 | 0.02119 |
| | $y_2$ | 0.030 | 0.001 | 0.0186 | 0.02092 |
| | $y_3$ | 0.030 | 0.001 | 0.0189 | 0.02110 |
| | $y_4$ | 0.033 | 0.003 | 0.0212 | 0.02299 |

**Table 13. The rubber allometric growth equation for above-ground and below-ground biomass.**

| Biomass | Allometric growth equation | Total biomass |
|---|---|---|
| Aboveground biomass | $W_T = 0.05712[(100D)^2H]^{0.94760}$ | $W = W_T + W_R$ |
| Underground biomass | $W_R = 0.0004406[(100D)^2H]^{1.18543}$ | |

* Note: D is DBH of trees , H is Tree height.

Using the optimal regression model, we estimated the DBH with the regression equation $y_3 = 0.01158x + 0.2156$, which was developed based on the E-W CD data. Due to the challenges in obtaining the specific carbon content of rubber trees, we used the average carbon content of 0.47, as recommended in the "Forest Carbon Stock Measurement and Monitoring Guidelines" for tropical and subtropical tree species, which is based on the average value suggested by the IPCC. This average value was used to convert biomass into carbon stock. The calculated carbon stock results for the artificial rubber forest are presented in Table 14.

The results presented in Table 8 indicate that the total aboveground biomass of the rubber plantations in the study area is 550,336.17 kg, while the total belowground bio-mass is 42,434.39 kg. The total biomass amounts to 592,770.57 kg, and the carbon stock is 278,602.17 kg.

## 5. Discussion

Forest carbon sequestration is recognized as a crucial strategy for mitigating greenhouse gas emissions and combating climate change. Accurate estimation of forest carbon stocks is essential for achieving global carbon neutrality targets. Traditional remote sensing techniques for large-scale carbon stock estimation face several challenges, including resolution limitations, incomplete data in complex terrains, reliance on model accuracy, and vulnerability to atmospheric interference. In contrast, LiDAR technology offers a significant advantage by capturing high-precision, three-dimensional point cloud data. This data enables precise measurement of tree height, canopy cover, and structural features. Unlike traditional methods, LiDAR operates independently of lighting and atmospheric conditions, making it effective in challenging environments such as dense forests and complex terrains. As a result, LiDAR provides valuable support for improving the accuracy of forest carbon stock estimation.

**Table 14. Calculation table for carbon stocks in rubber trees.**

| ID | H/m | E-W CBD(m) | D/m | WT/kg | WR/kg | W/kg | CS/kg |
|---|---|---|---|---|---|---|---|
| 1 | 20.5 | 4 | 0.186 | 254.45 | 16.17 | 270.62 | 127.19 |
| 2 | 20.9 | 5 | 0.232 | 395.57 | 28.08 | 423.65 | 199.12 |
| 3 | 21.1 | 4.3 | 0.200 | 299.92 | 19.86 | 319.78 | 150.30 |
| 4 | 21.7 | 6.8 | 0.316 | 734.12 | 60.86 | 794.98 | 373.64 |
| 5 | 22.1 | 6.5 | 0.302 | 685.72 | 55.88 | 741.60 | 348.55 |
| … | … | | … | … | … | … | … |
| 1316 | 19 | 6.3 | 0.293 | 560.05 | 43.38 | 603.42 | 283.61 |
| 1317 | 18.3 | 9.3 | 0.432 | 1130.67 | 104.46 | 1235.13 | 580.51 |
| 1318 | 19 | 5 | 0.232 | 361.41 | 25.08 | 386.49 | 181.65 |
| 1319 | 21.2 | 5.3 | 0.246 | 447.77 | 32.79 | 480.55 | 225.86 |
| 1320 | 20.1 | 6.3 | 0.293 | 590.73 | 46.37 | 637.10 | 299.44 |
| Sum | | | | 550336.17 | 42434.39 | 592770.57 | 278602.17 |

* Note: CS is carbon storage.

There are various methods for the rapid estimation of tree carbon stocks, with re-mote sensing traditionally being used for carbon stock estimation over large areas. While remote sensing techniques enable the acquisition of tree data over broad regions, they have limitations, such as resolution constraints, incomplete data in complex terrains, dependence on model accuracy, susceptibility to atmospheric interference, and the complexity of data processing. This study lever-ages drone-based LiDAR technology to rapidly acquire large-scale, high-precision 3D point cloud data of forest structure. Compared to remote sensing techniques, drone-based LiDAR has a clear advantage in resolution, enabling the accurate capture of key tree features, such as tree height, crown diameter, and trunk structure, thereby improving the accuracy of carbon stock estimation. Furthermore, LiDAR technology is unaffected by atmospheric and lighting conditions, allowing it to operate stably in complex terrains and dense forests, thus overcoming the data incompleteness issues often encoun-tered by remote sensing in these environments. Therefore, combining LiDAR technology with traditional remote sensing data can effectively enhance the precision and reliability of large-scale forest carbon stock estimation.

In this study, we compared three methods for segmenting individual trees: (1) CHM-based segmentation, (2) raw point cloud-based segmentation, and (3) seed point-based segmentation. We evaluated the performance of each method. Our comparative analysis showed that the CHM-based and seed point-based methods yielded better segmentation results under specific conditions. However, these methods often require intermediate steps, such as generating canopy mod-els or selecting seed points accurately, which limits their effectiveness in complex environments. In contrast, direct point cloud-based segmentation methods are more efficient in environments with dense tree cover, interlaced canopies, and complex terrain, as they can more accurately capture the three-dimensional structure of trees, providing higher segmen-tation precision [59]. We also explored individual tree segmentation using the PointNet++ model, based on deep learning. However, we encountered several issues, such as under-segmentation, over-segmentation, and missed segmentation [60]. Further analysis indicated that these problems were linked to factors such as the quality of airborne point cloud data, the complexity of tree structures, insufficient training data, and inadequate data preprocessing. As a result, the segmenta-tion outcomes from the PointNet++ model were not adopted. To improve the performance of deep learning models in tree segmentation, future research should focus on: (1) increasing data diversity by including point cloud data from various forest types, seasons, and environments, to enhance the model's ability to generalize [61]; and (2) optimizing the training process through network architecture improvements and the use of techniques such as multi-task learning and reinforce-ment learning to boost model robustness and accuracy [62].

Due to certain limitations in airborne LiDAR systems when acquiring point clouds of broadleaf trees, accurately cap-turing fine details of tree structure is challenging. This paper presents a method for estimating the DBH based on canopy diameter features. In order to accurately estimate the DBH of rubber trees, we constructed regression models, including power function, exponential function, linear function, and quadratic function, by analyzing three canopy diameter metrics: E-W CD, N-S CD, and A CD. These models were validated using error metrics such as maximum, minimum, average errors, and RMSE. The results showed that N-S CD was the most reliable independent variable, yielding the best per-formance across all indicators, particularly with small RMSE and average errors. Among the regression models based on N-S CD, the model $y_2$ (RMSE = 0.0225) exhibited the best performance. However, the extraction accuracy of N-S CD and A CD parameters did not meet the required standard, with only the E-W CD parameter meeting the computational conditions. Therefore, the regression model $y_3 = 0.01158x + 0.2156$, based on E-W CD, was chosen for DBH estimation. This method efficiently uses airborne LiDAR data and provides a fast, simple solution for estimating DBH across large forest areas. However, the method has some limitations: 1) the relationship between canopy diameter and DBH may differ across regions and tree species, affecting the model's generalizability; 2) estimation accuracy may be reduced in areas with small canopy widths or dense tree cover; 3) the quality of airborne LiDAR data in regions with overlapping tree crowns and complex terrain significantly impacts the model's accuracy.

The current UAV-based LiDAR technology primarily relies on vertical measurements, which often face challenges in penetrating densely vegetated areas, resulting in sparse point clouds. This limitation reduces the accuracy and reliability

 

of three-dimensional vegetation models, particularly in complex, multi-layered forest environments. Specifically, capturing structural details of understory and mid-story vegetation remains difficult, further complicating the identification of smaller tree features. As a result, the overall quality of the data is compromised. To address these issues, we propose an innovative data acquisition approach utilizing UAV-mounted LiDAR technology. This method aims to enhance the precision of vegetation point cloud models. By incorporating multi-angle, inclined flight paths—such as circular and grid-patterned routes for LiDAR tilt measurements—we can capture more comprehensive three-dimensional point cloud data of vegetation. This strategy improves the collection of data from varying heights and angles, offering robust support for the accurate construction of vegetation models and facilitating systematic analysis.

Regarding data acquisition, current UAV-borne LiDAR technology is widely used in forest resource surveys, forest management, and ecological research due to its efficient and high-precision data collection capabilities. It has become an essential tool in these areas. However, existing airborne LiDAR technology mainly relies on nadir measurements, which face challenges in penetrating dense forest canopies, often leading to sparse point clouds. This limitation affects the accuracy and reliability of three-dimensional vegetation data, especially in complex, multi-layered forest structures. The structural information of middle and lower vegetation layers is often difficult to capture, impacting the detection of small tree features and overall data quality. To address this, future research should focus on overcoming these limitations and developing more innovative data acquisition methods. One promising approach is the integration of airborne LiDAR with SLAM (Simultaneous Localization and Mapping) technology. This combination can improve positioning and mapping accuracy, enhancing the precision and comprehensiveness of data acquisition [63]. Another key area for future development is multi-source data fusion technology. By combining LiDAR data with high-definition imagery, photogrammetry, or other sensor data, the limitations of relying on a single data source can be mitigated, resulting in more accurate and comprehensive three-dimensional vegetation models [64]. In the next phase, we propose a new data acquisition strategy based on UAV-borne LiDAR technology to improve the accuracy of vegetation point cloud models. By employing multi-angle flight paths, such as circular and grid planning, we can achieve a more comprehensive collection of three-dimensional point cloud data. This method will improve the ability to collect data from various heights and angles, providing stronger support for constructing accurate vegetation point cloud models. It will also facilitate the systematic analysis of forest ecosystems, promoting the scientific development of forest ecosystem monitoring and management.

## 6. Conclusions

This paper presents a method for extracting rubber tree parameters and estimating carbon stock using airborne LiDAR data. Initially, we employed UAV-based LiDAR technology to efficiently and accurately capture large-scale, three-dimensional point cloud data of rubber plantations. To eliminate noise and classify ground points, Gaussian filtering and progressive triangulation mesh filtering algorithms were applied. Additionally, DEM, DSM and CHM were generated. Using the acquired data, we performed individual tree segmentation with four different approaches: segmentation based on the CHM, raw point cloud-based segmentation, seed point-based segmentation, and segmentation using the deep learning model PointNet++. The performance of these methods was evaluated. Among them, the raw point cloud-based segmentation demonstrated the best results in complex environments, accurately capturing the three-dimensional structure of trees with minimal dependence on environmental factors, making it more adaptable.

To overcome the challenge of missing DBH information in LiDAR point cloud data for rubber trees, we propose a new estimation method based on crown diameter features. This method focuses on analyzing the east-west crown diameter, north-south crown diameter, and the average crown diameter. We developed regression models using various functions, including power, exponential, linear, and quadratic functions. These models were validated using error metrics such as maximum error, minimum error, mean error, and RMSE. Among the models, the one based on the E-W CD ($y_3 = 0.01158x + 0.2156$) provided the best fit and was selected for further estimation of carbon storage. This model compensates for the lack of DBH data in the LiDAR point cloud. Finally, we used the allometric growth equation for rubber

trees and the average carbon content parameter to calculate the biomass and carbon storage in the study area's rubber forests. This approach offers an effective method for large-scale carbon sink assessment using LiDAR data.

In future studies, we will conduct validation and comparative tests on various rubber tree species to assess the model's generalizability and robustness. Additionally, by integrating multi-source remote sensing data, such as spectral information and ground-based LiDAR measurements, and applying advanced techniques like deep learning, we aim to improve the accuracy and adaptability of individual tree segmentation and DBH estimation. These improvements will contribute significantly to the broader estimation of forest carbon stocks.

## Supporting information

**S1 File. Point cloud data.**
(RAR)

## Author contributions

**Conceptualization:** HaoYu Tai, Xia Li.

**Data curation:** Hongen Li.

**Formal analysis:** Chuangjiang Rao.

**Project administration:** Xia Li.

**Resources:** Chuangjiang Rao, Hongen Li.

**Validation:** HaoYu Tai, Chen Li.

**Writing – original draft:** HaoYu Tai.

**Writing – review & editing:** Xia Li.

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
