## [Decision Letter · Decision Letter 0]

10 Mar 2025

PONE-D-24-57180Methods for Extracting Rubber Tree Parameters and Estimating Carbon Storage Using Airborne LiDAR Point Cloud DataPLOS ONE

Dear Dr. Rao,

Thank you for submitting your manuscript to PLOS ONE. After careful consideration, we feel that it has merit but does not fully meet PLOS ONE’s publication criteria as it currently stands. Therefore, we invite you to submit a revised version of the manuscript that addresses the points raised during the review process.

The manuscript must be corrected in all points indicated by the reviewers, such as:

1. The title is inappropriate as it lacks an academic summary of the work. A mere introduction of the methodology fails to effectively encapsulate the thesis's theme.

2. The introduction lacks a clear statement of the research objectives and further clarification is needed regarding the editorial contributions of this work.

3. The authors seem to have overlooked the comparison of their method with others, particularly in the discussion section, where a comprehensive discussion on the differences in accuracy, applicability, and other aspects of this work relative to other methods is missing.

4. Presenting part of the content in Section 4.3 in a tabular form would be more effective.

5. The conclusion section is incomplete. While the authors mention an accuracy analysis, they fail to report the results of this analysis.

6. The authors state in the conclusion that their method "provides a reliable solution," which lacks factual support. At least one corresponding validation of accuracy or applicability is required.

7. The proposed method for estimating Diameter at Breast Height (DBH) using the east-west crown diameter is innovative. However, it is unclear how significantly it improves upon existing DBH estimation methods.

8. The study evaluates three segmentation techniques, ultimately selecting the best-performing method. The paper does not convincingly demonstrate a groundbreaking advancement over previous work.

9.The use of LiDAR for biomass and carbon storage estimation is well established. The novelty lies more in its specific adaptation to rubber trees rather than in a fundamentally new approach.

10. Some sentences are overly complex, making the paper difficult to read. For instance, the abstract and introduction contain long, dense sentences that could be more concise.

11. Some descriptions are unnecessarily repetitive or overly detailed. The methodology section, for example, includes long explanations of LiDAR operation that may not be necessary for the intended audience.

12. While not abundant, some minor grammatical inconsistencies and awkward phrasing are present.

We look forward to receiving your revised manuscript.

Kind regards,

Claudionor Ribeiro da Silva

Academic Editor

PLOS ONE

 [This work was financially supported by the National Natural Science Foundation of China (Grant Nos. 62266026); the High Score Special Project: Yunnan Provincial Government Comprehensive Governance Deep Application and Scale-up Industrialization Demonstration Project. (Grant Nos. 89-Y50G31-9001-22/23).]. 

Please state what role the funders took in the study.  If the funders had no role, please state: ""The funders had no role in study design, data collection and analysis, decision to publish, or preparation of the manuscript.

5. We note that your Data Availability Statement is currently as follows: [All relevant data are within the manuscript and its Supporting Information files.]

6. We note that Figure 2 in your submission contain [map/satellite] images which may be copyrighted. All PLOS content is published under the Creative Commons Attribution License (CC BY 4.0), which means that the manuscript, images, and Supporting Information files will be freely available online, and any third party is permitted to access, download, copy, distribute, and use these materials in any way, even commercially, with proper attribution. For these reasons, we cannot publish previously copyrighted maps or satellite images created using proprietary data, such as Google software (Google Maps, Street View, and Earth). For more information, see our copyright guidelines: http://journals.plos.org/plosone/s/licenses-and-copyright.

A. You may seek permission from the original copyright holder of Figure 2 to publish the content specifically under the CC BY 4.0 license. 

B. If you are unable to obtain permission from the original copyright holder to publish these figures under the CC BY 4.0 license or if the copyright holder’s requirements are incompatible with the CC BY 4.0 license, please either i) remove the figure or ii) supply a replacement figure that complies with the CC BY 4.0 license. Please check copyright information on all replacement figures and update the figure caption with source information. If applicable, please specify in the figure caption text when a figure is similar but not identical to the original image and is therefore for illustrative purposes only.

7. Please upload a copy of Figure 9, to which you refer in your text on page 12. If the figure is no longer to be included as part of the submission please remove all reference to it within the text.

Reviewers' comments:

Reviewer's Responses to Questions

**Comments to the Author**

1. Is the manuscript technically sound, and do the data support the conclusions?

Reviewer #1: Yes

Reviewer #2: Yes

2. Has the statistical analysis been performed appropriately and rigorously? 

Reviewer #1: Yes

Reviewer #2: Yes

3. Have the authors made all data underlying the findings in their manuscript fully available?

Reviewer #1: Yes

Reviewer #2: Yes

4. Is the manuscript presented in an intelligible fashion and written in standard English?

Reviewer #1: Yes

Reviewer #2: Yes

5. Review Comments to the Author

Reviewer #1: 1. The title is inappropriate as it lacks an academic summary of the work. A mere introduction of the methodology fails to effectively encapsulate the thesis's theme.

2. The introduction lacks a clear statement of the research objectives and further clarification is needed regarding the editorial contributions of this work.

3. The authors seem to have overlooked the comparison of their method with others, particularly in the discussion section, where a comprehensive discussion on the differences in accuracy, applicability, and other aspects of this work relative to other methods is missing.

4. Presenting part of the content in Section 4.3 in a tabular form would be more effective.

5. The conclusion section is incomplete. While the authors mention an accuracy analysis, they fail to report the results of this analysis.

6. The authors state in the conclusion that their method "provides a reliable solution," which lacks factual support. At least one corresponding validation of accuracy or applicability is required. If the authors wish to prove the reliability of their results, they need to conduct repeated experiments in at least three different applicable regions, as a single experiment cannot account for the impact of randomness.

Reviewer #2: The paper may be considered for publication upon the incorporation of the following comments and suggestions.

Novelty Assessment:

The paper presents a method for extracting rubber tree parameters and estimating carbon storage using airborne LiDAR. The study has some novel aspects, but its originality could be questioned in certain areas:

1. DBH Estimation Method: The proposed method for estimating Diameter at Breast Height (DBH) using the east-west crown diameter is innovative. However, it is unclear how significantly it improves upon existing DBH estimation methods. A comparison with alternative regression models or machine learning-based approaches would strengthen the claim of novelty.

2. LiDAR-Based Tree Segmentation: The study evaluates three segmentation techniques (CHM-based, direct point cloud-based, and seed point-based), ultimately selecting the best-performing method. While this is valuable, segmentation methods for tree structure analysis have been widely studied. The paper does not convincingly demonstrate a groundbreaking advancement over previous work.

3. Carbon Storage Estimation: The use of LiDAR for biomass and carbon storage estimation is well established. The study provides an application in rubber plantations, but similar methods have been applied to other forest types. The novelty lies more in its specific adaptation to rubber trees rather than in a fundamentally new approach.

Academic English Assessment:

1. Some sentences are overly complex, making the paper difficult to read. For instance, the abstract and introduction contain long, dense sentences that could be more concise. Breaking complex ideas into smaller, more digestible parts would improve readability.

2. While technical terminology is expected in a scientific paper, some descriptions are unnecessarily repetitive or overly detailed. The methodology section, for example, includes long explanations of LiDAR operation that may not be necessary for the intended audience.

3. Grammar & Stylistic Issues: While not abundant, some minor grammatical inconsistencies and awkward phrasing are present. Examples include:

"Given that airborne LiDAR technology cannot directly obtain the diameter at breast height (DBH) parameters of rubber forests..." → This sentence could be more direct: "Since airborne LiDAR cannot directly measure DBH in rubber forests..."

"The study results indicate: (1) The BB4 quad-copter drone equipped with the AU20 LiDAR measurement system can rapidly acquire point cloud data from artificial rubber forests..." → Listing results with numbers in the abstract is not ideal for readability.

Section wise comments:

Title

The title is clear and informative, but it could be more concise. Consider simplifying it to "Extracting Rubber Tree Parameters and Estimating Carbon Storage Using Airborne LiDAR."

The phrase "Methods for Extracting" might be unnecessary, as the study itself presents a specific approach.

Abstract:

Consider simplifying technical jargon to enhance readability.

A brief mention of potential limitations or future applications could strengthen the conclusion.

Instead of listing results in a numbered format, integrate them into sentences for smoother readability.

Introduction:

The introduction provides a strong justification for the study by linking it to global challenges. However, the transition from global climate issues to LiDAR technology could be smoother.

More discussion on why previous methods are insufficient would strengthen the rationale for the study.

Literature Review

The discussion of past research is thorough, but the review would benefit from a clearer structure. For example, separating literature into subtopics (e.g., carbon sequestration methods, LiDAR in forestry, DBH estimation techniques) could improve readability.

It would be useful to highlight more gaps in existing studies to further justify the research contribution.

Methodology:

The methodology is detailed and technically sound, but a schematic diagram outlining the workflow could improve clarity.

The description of UAV and LiDAR specifications is useful, but some sections (e.g., details about software and settings) may be overly detailed for a general audience. Consider summarizing technical parameters in a table.

It would be helpful to justify why three methods were chosen and whether hybrid approaches were considered.

Results

Some figures and tables could be better formatted to emphasize key comparisons.

A discussion on why the best-performing method outperformed others would add value.

The estimation of DBH using crown diameter is a novel contribution, but potential sources of error or limitations in different forest conditions should be acknowledged.

Discussion:

The discussion effectively contextualizes the results, but it could better compare findings with previous research. Were similar DBH estimation approaches used elsewhere?

The limitations of airborne LiDAR are noted, but suggestions for overcoming them (e.g., combining LiDAR with ground-truthing) is essential.

Future research directions are mentioned, but expanding on how machine learning or AI could enhance segmentation would be insightful.

Conclusion

The conclusion effectively summarizes findings, but it could better emphasize the practical applications of the study. How can this method be scaled for different regions?

The impact of the study on carbon sequestration policy or rubber plantation management could be further highlighted.

A brief discussion on potential next steps (e.g., testing the model in different forest types) would add depth.

References:

Incorporating recent references, particularly those related to machine learning-based studies, is essential for providing readers with a better understanding of current research trends."

Formatting should be double-checked for consistency, especially regarding journal abbreviations and citation styles.

The paper requires substantial revisions in the following areas.

1. Clarify Novelty: The paper presents incremental improvements rather than a fundamentally new approach. The novelty claim needs to be reinforced by comparing the proposed DBH estimation method with other regression-based or machine-learning models. Otherwise, the study risks being viewed as a case study rather than a significant scientific breakthrough.

2. Strengthen the Literature Review: The paper discusses previous work but does not sufficiently highlight how this study differs from existing methods. What specific gaps does this study fill? If similar segmentation methods have been applied in other forestry contexts, why is this study different?

3. Improve Readability & Academic Tone: The paper should undergo careful revision to simplify overly complex sentences, remove redundant explanations, and improve clarity. This will enhance accessibility without compromising technical depth.

4. Address Limitations More Explicitly: The discussion acknowledges the limitations of LiDAR, but the paper should go further by suggesting specific solutions. How can occlusion issues be mitigated? Would integrating multispectral or hyperspectral data improve results? How does tree density affect segmentation accuracy?

5. Provide a More Convincing Justification for Method Selection: The study evaluates three segmentation methods but does not clearly justify why these three were chosen over others. Were deep learning-based segmentation methods considered? If not, why?

Final Verdict

This study presents a valuable and technically rigorous contribution to LiDAR-based forestry analysis.

While the study has merit, its novelty is moderate rather than groundbreaking. The paper is written in standard academic English, but its readability could be improved. Strengthening the novelty claim, refining the discussion, and improving writing clarity will significantly enhance its impact and publication potential.

6. PLOS authors have the option to publish the peer review history of their article (what does this mean? ). If published, this will include your full peer review and any attached files.

**Do you want your identity to be public for this peer review?** For information about this choice, including consent withdrawal, please see our Privacy Policy .

Reviewer #1: No

Reviewer #2: **Yes: ** Dr. A Salim Khan

---

## [Author Response · Author response to Decision Letter 1]

16 Apr 2025

Response to Reviewers

Dear Reviewers,

We deeply appreciate the effort and time that you have spent in reviewing our manuscript “Extracting Rubber Tree Parameters and Estimating Carbon Storage Using Airborne LiDAR” (ID: PONE-D-24-57180). We have studied the comments carefully and have made revisions which we hope meet with your approval. The revised outline is as follows:

We will add some quantitative results to the abstract section after the experiment. (Please refer to the abstract section)

We have added a discussion on the application of laser technology in the introduction section. (Please refer to the introduction section)

We have added some discussions on single tree segmentation methods based on deep learning methods. (Please refer to the section on single tree segmentation methods)

We have redesigned the method for estimating DBH. (Please refer to the DBH estimation method section in the methodology.)

We have made detailed revisions and expansions to the abstract, introduction, methodology, discussion, and conclusion.

We carefully checked the experimental results of the entire manuscript.

We carefully checked and modified the image format of the entire manuscript.

We have revised the entire manuscript and carefully addressed some terminology issues.

Response to reviewer 1 comments

Point 1: The title is inappropriate as it lacks an academic summary of the work. A mere introduction of the methodology fails to effectively encapsulate the thesis's theme.

Response 1: Thank you for helpful comment. We have revised the title of the article and adopted the comments of the reviewers.

Point 2: The introduction lacks a clear statement of the research objectives and further clarification is needed regarding the editorial contributions of this work.

Response 2: Thank you for helpful comment. We have made detailed revisions to the introduction. Please refer to the revised manuscript's introduction section, pages 2 to 5.

Point 3: The authors seem to have overlooked the comparison of their method with others, particularly in the discussion section, where a comprehensive discussion on the differences in accuracy, applicability, and other aspects of this work relative to other methods is missing.

Response 3: Thank you for helpful comment. We have added a deep learning based single tree segmentation method to our approach and conducted an analysis on it. For the method of estimating breast diameter parameters, we redesigned the experiment and constructed regression models such as power function, exponential function, linear function, and quadratic function by analyzing the parameters of east-west crown diameter, north-south crown diameter, and average crown diameter. These models were validated using indicators such as maximum, minimum, average error, and RMSE. In the discussion section, we have made modifications and expanded the methods. Please refer to section 3.4.3 of the paper, pages 13 to 14. Section 4.3, pages 18 to 21.

Point 4: Presenting part of the content in Section 4.3 in a tabular form would be more effective.

Response 4: Thank you for helpful comment. We have made detailed modifications to section 4.3 and presented it in the form of a table. Please refer to section 4.3, pages 18 to 21.

Point 5: The conclusion section is incomplete. While the authors mention an accuracy analysis, they fail to report the results of this analysis.

Response 5: Thank you for helpful comment. We have made revisions to the conclusion section. Please refer to the conclusion section, pages 26 to 27.

Point 6: The authors state in the conclusion that their method "provides a reliable solution," which lacks factual support. At least one corresponding validation of accuracy or applicability is required. If the authors wish to prove the reliability of their results, they need to conduct repeated experiments in at least three different applicable regions, as a single experiment cannot account for the impact of randomness.

Response 6: Thank you for helpful comment. We have made modifications to the method and discussed it. We have made revisions to the conclusion section. Please refer to the conclusion section, pages 26 to 27.

Response to reviewer 2 comments

Point 1: Clarify Novelty: The paper presents incremental improvements rather than a fundamentally new approach. The novelty claim needs to be reinforced by comparing the proposed DBH estimation method with other regression-based or machine-learning models. Otherwise, the study risks being viewed as a case study rather than a significant scientific breakthrough.

Response 1: Thank you for helpful comment. For the method of estimating breast height diameter parameters, we redesigned the experiment and constructed regression models such as power function, exponential function, linear function, and quadratic function by analyzing the parameters of east-west crown diameter, north-south crown diameter, and average crown diameter. These models were validated using metrics such as maximum, minimum, average error, and RMSE. In the discussion section, we made modifications and expanded the methods. Please refer to Section 4.3, pages 18-21 of the paper.

Point 2: Strengthen the Literature Review: The paper discusses previous work but does not sufficiently highlight how this study differs from existing methods. What specific gaps does this study fill? If similar segmentation methods have been applied in other forestry contexts, why is this study different?

Response 2: Thank you for helpful comment. We have made revisions to the introduction, methods, and discussion sections. We have strengthened the literature review and discussed our methods.

Point 3: Improve Readability & Academic Tone: The paper should undergo careful revision to simplify overly complex sentences, remove redundant explanations, and improve clarity. This will enhance accessibility without compromising technical depth.

Response 3: Thank you for helpful comment. We carefully revised the entire manuscript and modified some terminology. And simplified some complex sentences.

Point 4: Address Limitations More Explicitly: The discussion acknowledges the limitations of LiDAR, but the paper should go further by suggesting specific solutions. How can occlusion issues be mitigated? Would integrating multispectral or hyperspectral data improve results? How does tree density affect segmentation accuracy?

Response 4: Thank you for helpful comment. We have made modifications in the discussion section, discussing the limitations of lidar and providing our next research direction. Please refer to the discussion section.

Point 5: Provide a More Convincing Justification for Method Selection: The study evaluates three segmentation methods but does not clearly justify why these three were chosen over others. Were deep learning-based segmentation methods considered? If not, why?

Response 5: Thank you for helpful comment. We have added experiments based on deep learning to the single tree segmentation method and discussed its practicality. Please refer to section 3.4.3, pages 13 to 14, and the discussion section.

Response to reviewer 3 comments

Point 1: Please ensure that your manuscript meets PLOS ONE's style requirements, including those for file naming.

Response 1: Thank you for helpful comment. We have made modifications to the article according to the template, and the file name has also been changed.

Point 2: In your Methods section, please provide additional information regarding the permits you obtained for the work. Please ensure you have included the full name of the authority that approved the field site access and, if no permits were required, a brief statement explaining why.

Response 2: Thank you for helpful comment. We have added project sources in the methodology. Please refer to the overview section of the research area in the methodology, page 5.

Point 3: Please state what role the funders took in the study. If the funders had no role, please state: ""The funders had no role in study design, data collection and analysis, decision to publish, or preparation of the manuscript.

Response 3: Thank you for helpful comment. We have made modifications and added this explanation.

Point 4: We note that Figure 2 in your submission contain [map/satellite] images which may be copyrighted.

Response 4: Thank you for helpful comment. We have made modifications to Figure 2 and used our own orthorectified images as map data.

Point 5: Please upload a copy of Figure 9, to which you refer in your text on page 12. If the figure is no longer to be included as part of the submission please remove all reference to it within the text.

Response 5: Thank you for helpful comment. We have made modifications to Figure 9, please refer to the Methods section on page 11.

---

## [Decision Letter · Decision Letter 1]

6 Jun 2025

PONE-D-24-57180R1Extracting Rubber Tree Parameters and Estimating Carbon Storage Using Airborne LiDARPLOS ONE

Dear Dr. Rao,

Thank you for submitting your manuscript to PLOS ONE. After careful consideration, we feel that it has merit but does not fully meet PLOS ONE’s publication criteria as it currently stands. Therefore, we invite you to submit a revised version of the manuscript that addresses the points raised during the review process.

We look forward to receiving your revised manuscript.

Kind regards,

Claudionor Ribeiro da Silva

Academic Editor

PLOS ONE

Journal Requirements:

Reviewers' comments:

Reviewer's Responses to Questions

**Comments to the Author**

1. If the authors have adequately addressed your comments raised in a previous round of review and you feel that this manuscript is now acceptable for publication, you may indicate that here to bypass the “Comments to the Author” section, enter your conflict of interest statement in the “Confidential to Editor” section, and submit your "Accept" recommendation.

Reviewer #1: All comments have been addressed

Reviewer #3: (No Response)

2. Is the manuscript technically sound, and do the data support the conclusions?

Reviewer #1: Yes

Reviewer #3: Yes

3. Has the statistical analysis been performed appropriately and rigorously? 

Reviewer #1: Yes

Reviewer #3: Yes

4. Have the authors made all data underlying the findings in their manuscript fully available?

Reviewer #1: Yes

Reviewer #3: Yes

5. Is the manuscript presented in an intelligible fashion and written in standard English?

Reviewer #1: Yes

Reviewer #3: Yes

6. Review Comments to the Author

Reviewer #1: (No Response)

Reviewer #3: Abstract

1) Please make the introductory section of the abstract concise, it should be summarized in 3-4 lines

2) The abstracts section is a much longer, need to shorter

3) The abstract much cover the introduction and problem statement in 4-5 lines, the methodology (2-3 lines), major results/findings and discussion (4-6 lines) and importance/implication/application/conclusion (2-3 lines).

Introduction

1) Please include some numerical data with respect forest and climate change the following articles might be helpful

2) Yin, Y., Gong, H., Chen, Z., Tian, X., Wang, Y., Wang, Z.,... Cui, Z. (2025). Underestimated sequestration of soil organic carbon in China. Environmental Chemistry Letters, 23(2), 373-379. doi: 10.1007/s10311-024-01813-4

Adnan Ahmad1†, Shahid Ahmad4†, Ghulam Nabi*, Alam Zeb2, Muhammad Nawaz Rajpar2, Sami Ullah2, Faisal Khalid1,2, Mujibur Rahman5, Qijing Liu 3, Kuiling Zu6, Xinle Guo and Kunyuan Wanghe (2022). Carbon Emissions With Forest Cover Change and Wood Harvest in the Dry Temperate Region of Pakistan Between 1908 and 2015: Front. Environ. Sci. 10:876225. doi: 10.3389/fenvs.2022.876225

3) Please include some numerical information on the status of forest carbon sequestration at global as well as China

4) Pleases enrich the introduction section with updated literature that why the use of technology and computing is needed in forest measurement. The following article might me helpful

Ma, X., Ding, J., Wang, T., Lu, L., Sun, H., Zhang, F.,... Nurmemet, I. (2023). A Pixel Dichotomy Coupled Linear Kernel-Driven Model for Estimating Fractional Vegetation Cover in Arid Areas From High-Spatial-Resolution Images. IEEE Transactions on Geoscience and Remote Sensing, 61. doi: 10.1109/TGRS.2023.3289093.

Sun, H., Ma, X., Liu, Y., Zhou, G., Ding, J., Lu, L.,... Zhang, F. (2024). A New Multiangle Method for Estimating Fractional Biocrust Coverage From Sentinel-2 Data in Arid Areas. IEEE Transactions on Geoscience and Remote Sensing, 62, 1-15. doi: 10.1109/TGRS.2024.3361249

Zhou, G., Zhou, X., Li, W., Zhao, D., Song, B., Xu, C.,... Zou, L. (2022). Development of a Lightweight Single-Band Bathymetric LiDAR. Remote Sensing, 14(22), 5880. doi: 10.3390/rs14225880

Wang, B., Yang, M., Cao, P., & Liu, Y. (2025). A novel embedded cross framework for high-resolution salient object detection. Applied Intelligence, 55(4), 277. doi: 10.1007/s10489-024-06073-x

Zhou, G., Xu, J., Hu, H., Liu, Z., Zhang, H., Xu, C.,... Zhao, Y. (2023). Off-Axis Four-Reflection Optical Structure for Lightweight Single-Band Bathymetric LiDAR. IEEE Transactions on Geoscience and Remote Sensing, 61. doi: 10.1109/TGRS.2023.3298531

Li, R., Wang, Y., Sun, S., Zhang, Y., Ding, F.,... Gao, H. (2025). UE-Extractor: A Grid-to-Point Ground Extraction Framework for Unstructured Environments Using Adaptive Grid Projection. IEEE Robotics and Automation Letters, 10(6), 5991-5998. doi: 10.1109/LRA.2025.3563127

5) Problem statement and justification section need modification supported by literature

Discussion and conclusion

1) The discussion section need modification, the statement that belongs to author must be supported by the empirical data and those does not belong to author, a sources of citation might be provided. Also how the present findings are consistent or inconsistent with other must be provided in a logical, coherent and connected way . The following articles might be useful

Ma, X., Ding, J., Wang, T., Lu, L., Sun, H., Zhang, F.,... Nurmemet, I. (2023). A Pixel Dichotomy Coupled Linear Kernel-Driven Model for Estimating Fractional Vegetation Cover in Arid Areas From High-Spatial-Resolution Images. IEEE Transactions on Geoscience and Remote Sensing, 61. doi: 10.1109/TGRS.2023.3289093.

Sun, H., Ma, X., Liu, Y., Zhou, G., Ding, J., Lu, L.,... Zhang, F. (2024). A New Multiangle Method for Estimating Fractional Biocrust Coverage From Sentinel-2 Data in Arid Areas. IEEE Transactions on Geoscience and Remote Sensing, 62, 1-15. doi: 10.1109/TGRS.2024.3361249

Zhou, G., Zhou, X., Li, W., Zhao, D., Song, B., Xu, C.,... Zou, L. (2022). Development of a Lightweight Single-Band Bathymetric LiDAR. Remote Sensing, 14(22), 5880. doi: 10.3390/rs14225880

Wang, B., Yang, M., Cao, P., & Liu, Y. (2025). A novel embedded cross framework for high-resolution salient object detection. Applied Intelligence, 55(4), 277. doi: 10.1007/s10489-024-06073-x

Zhou, G., Xu, J., Hu, H., Liu, Z., Zhang, H., Xu, C.,... Zhao, Y. (2023). Off-Axis Four-Reflection Optical Structure for Lightweight Single-Band Bathymetric LiDAR. IEEE Transactions on Geoscience and Remote Sensing, 61. doi: 10.1109/TGRS.2023.3298531

Li, R., Wang, Y., Sun, S., Zhang, Y., Ding, F.,... Gao, H. (2025). UE-Extractor: A Grid-to-Point Ground Extraction Framework for Unstructured Environments Using Adaptive Grid Projection. IEEE Robotics and Automation Letters, 10(6), 5991-5998. doi: 10.1109/LRA.2025.3563127

2) Pleases include a limitation section in conclusion, also provide the importance of your study

7. PLOS authors have the option to publish the peer review history of their article (what does this mean? ). If published, this will include your full peer review and any attached files.

**Do you want your identity to be public for this peer review?** For information about this choice, including consent withdrawal, please see our Privacy Policy .

Reviewer #1: No

Reviewer #3: **Yes: ** Adnan Ahmad

---

## [Author Response · Author response to Decision Letter 2]

10 Jul 2025

Response to Reviewers

Dear Reviewers,

We deeply appreciate the effort and time that you have spent in reviewing our manuscript “Extracting Rubber Tree Parameters and Estimating Carbon Storage Using Airborne LiDAR” (ID: PONE-D-24-57180). We have studied the comments carefully and have made revisions which we hope meet with your approval. The revised outline is as follows:

We have made detailed revisions and expansions to the abstract, introduction, methodology, discussion, and conclusion.

We carefully checked the experimental results of the entire manuscript.

We carefully checked and modified the image format of the entire manuscript.

We have revised the entire manuscript and carefully addressed some terminology issues.

Response to reviewer 3 comments

Point 1: Abstract

1) Please make the introductory section of the abstract concise, it should be summarized in 3-4 lines

2) The abstracts section is a much longer, need to shorter

3) The abstract much cover the introduction and problem statement in 4-5 lines, the methodology (2-3 lines), major results/findings and discussion (4-6 lines) and importance/implication/application/conclusion (2-3 lines).

Response 1: Thank you for helpful comment. We have made revisions to the abstract, please refer to the abstract section for details, page 2.

Point 2: Introduction

1) Please include some numerical data with respect forest and climate change the following articles might be helpful.

3) Please include some numerical information on the status of forest carbon sequestration at global as well as China

4) Pleases enrich the introduction section with updated literature that why the use of technology and computing is needed in forest measurement.

5) Problem statement and justification section need modification supported by literature

Response 2: Thank you for helpful comment. We have made overall revisions to the introduction section. Added numerical information on forests and climate change, global carbon storage, and carbon neutrality. We explained from the perspective of forestry carbon sinks why technology and calculations are needed in forest measurement, and cited some references, pages 2 to 6.

Point 3: Discussion and conclusion

1) The discussion section need modification, the statement that belongs to author must be supported by the empirical data and those does not belong to author, a sources of citation might be provided. Also how the present findings are consistent or inconsistent with other must be provided in a logical, coherent and connected way.

2) Pleases include a limitation section in conclusion, also provide the importance of your study.

Response 3: Thank you for helpful comment. We have made modifications to the discussion and conclusion sections, pages 23 to 27.

---

## [Decision Letter · Decision Letter 2]

6 Aug 2025

Extracting Rubber Tree Parameters and Estimating Carbon Storage Using Airborne LiDAR

PONE-D-24-57180R2

Dear Dr. Rao,

We’re pleased to inform you that your manuscript has been judged scientifically suitable for publication and will be formally accepted for publication once it meets all outstanding technical requirements.

Kind regards,

Claudionor Ribeiro da Silva

Academic Editor

PLOS ONE

Additional Editor Comments (optional):

Reviewers' comments:

Reviewer's Responses to Questions

**Comments to the Author**

1. If the authors have adequately addressed your comments raised in a previous round of review and you feel that this manuscript is now acceptable for publication, you may indicate that here to bypass the “Comments to the Author” section, enter your conflict of interest statement in the “Confidential to Editor” section, and submit your "Accept" recommendation.

Reviewer #1: All comments have been addressed

Reviewer #3: All comments have been addressed

2. Is the manuscript technically sound, and do the data support the conclusions?

Reviewer #1: Yes

Reviewer #3: Yes

3. Has the statistical analysis been performed appropriately and rigorously? 

Reviewer #1: Yes

Reviewer #3: Yes

4. Have the authors made all data underlying the findings in their manuscript fully available?

Reviewer #1: Yes

Reviewer #3: Yes

5. Is the manuscript presented in an intelligible fashion and written in standard English?

Reviewer #1: Yes

Reviewer #3: Yes

6. Review Comments to the Author

Reviewer #1: Thank the author team for their revisions and clarifications to the paper. The current manuscript is acceptable for publication.

Reviewer #3: The MS isuch improved, please have a look carefully on spelling and Grammer mistakes , also please croesscheck the citation and references

7. PLOS authors have the option to publish the peer review history of their article (what does this mean? ). If published, this will include your full peer review and any attached files.

**Do you want your identity to be public for this peer review?** For information about this choice, including consent withdrawal, please see our Privacy Policy .

Reviewer #1: No

Reviewer #3: No

---

## [Editor Report · Acceptance letter]

PONE-D-24-57180R2

PLOS ONE

Dear Dr. Rao,

I'm pleased to inform you that your manuscript has been deemed suitable for publication in PLOS ONE. Congratulations! Your manuscript is now being handed over to our production team.

Kind regards,

on behalf of

Dr. Claudionor Ribeiro da Silva

Academic Editor

PLOS ONE